# Influence of manganese cycling on alkalinity in the redox stratified water column of Chesapeake Bay

Aubin Thibault de Chanvalon[1,2], George W. Luther[2], Emily R. Estes[2,*], Jennifer Necker[2], Bradley M. Tebo[3], Jianzhong Su[4,+], Wei-Jun Cai[4]

[1]Université de Pau et des Pays de l'Adour, E2S UPPA, CNRS, IPREM, Pau, France.
[2]School of Marine Science and Policy, University of Delaware, Lewes, Delaware, 19958 USA
[3]Division of Environmental and Biomolecular Systems, 3181 SW Sam Jackson Park Road, Portland, OR 97239, USA; Current address: Department of Chemistry, University of Washington, Seattle,WA 98195-1700
[4]School of Marine Science and Policy, University of Delaware, Newark, Delaware, 19716 USA
[+]State Key Laboratory of Marine Resources Utilization in South China Sea, Hainan University, Haikou, China
[*]Texas A&M University, USA

*Correspondence to*: A. Thibault de Chanvalon (aubin.thibault-de-chanvalon@univ-pau.fr)

Abstract.

The alkalinity dynamics in coastal environments play a crucial role in controlling the global burial of carbonate minerals
and the ocean's capacity to sequester anthropogenic $CO_2$. This study presents results from high vertical resolution profiles obtained during two summers in the temperate Chesapeake Bay estuary, enabling detailed investigation of carbonate dynamics over salinity and redox gradients, along with measurement of the speciation of most redox-sensitive elements. Under oxygen-rich conditions, carbonate dissolution, primary production, and aerobic respiration explain the evolution of total alkalinity (TA) versus dissolved inorganic carbon (DIC), once adjusted for fresh and oceanic water mixing. A significant flooding event
in 2018 promoted carbonate dissolution. In oxygen-depleted waters, we observed a previously unreported 2.4-mole increase in DIC per 1 mole of TA production, which was consistent over the two years. Stoichiometric changes suggest that $MnO_2$ reduction followed by Mn carbonate precipitation is responsible for this characteristic carbonate signature, likely produced in sediment pore water and then transferred to the water column along with other by-products of anoxic respiration at the onset of summer. Our findings highlight the critical role of Mn in alkalinity dynamics in the Chesapeake Bay and potentially other
river-dominated environments where it can limit $H_2S$ oxidation to $SO_4^{2-}$ and promote Sulphur burial.

# 1 Introduction

About 30% of anthropogenic $CO_2$ emissions are rapidly trapped by dissolution in the ocean as dissolved inorganic carbon (DIC), which is dominated by bicarbonate ions ($HCO_3^-$, Friedlingstein et al., 2019). At the century time scale, atmosphere-ocean exchanges result in oceanic $HCO_3^-$ enrichment not associated with a cationic enrichment, as measured by total alkalinity (TA). This is in contrast to silicate or carbonate weathering, which preponderates at timescales of thousands to millions of years and increases TA simultaneously with DIC (Urey, 1952). This cationic deficiency corresponds to an excess of proton release that can be tracked by an indicator such as TA minus DIC (Xue and Cai, 2020) and informs about the reduction of $CO_2$ buffering capacity of seawater (Gustafsson et al., 2019). The proton excess is only balanced in the deep ocean by the production of $Ca^{2+}$ from carbonate dissolution, a process named chemical carbonate compensation (Boudreau et al., 2018). However, in shallow waters, that accounts for 2/3 of global buried carbonate (Smith and Mackenzie, 2016), carbonate precipitation largely predominates over dissolution and other localised processes may constrain carbonate dynamics (Borges et al., 2006; Lohrenz et al., 2010). For example, some calcifying species may slow down their carbonate precipitation in case of pH decrease (so called biological carbonate compensation; Boudreau et al., 2018). As another example, in estuaries, the seasonality of river flow, temperature and continental erosion modulates $CaCO_3$ dissolution (e.g. Su et al., 2020b), atmospheric $CO_2$ exchanges (e.g. Borges et al., 2018), respiratory activity (e.g. Abril et al., 2003) and transfer of carbonate particles (Meybeck, 1987; Middelburg et al., 2020).

The global trends of human migration towards littoral areas and global warming favour eutrophication and a decrease in oxygen levels in coastal water (Breitburg et al., 2018; Rabalais et al., 2014). Sometimes driven by recurrent natural process (e.g. Gupta et al., 2021), exceptional events (e.g. Hulot et al., 2023), stratification reinforced by global warming (e.g. Meire et al., 2013) or anthropogenic nutrient loading (e.g. Hagy et al., 2004; Carstensen et al., 2018); coastal deoxygenation enhanced the possible build-up of anoxic conditions to build up as a permanent or seasonal feature. In anoxic environments, the drivers of the carbonate cycle change as the terminal electron acceptor for organic matter remineralisation shifts from $O_2$ to Mn oxides, Fe oxides then sulphate. Notably, the consumption of sulphate, followed by sulphur burial, produces an significant alkalinity flux toward coastal waters, accounting for approximately 10 % of the global alkalinity input to the ocean (Middelburg et al., 2020). Fe and Mn oxides play a critical role in sulphur burial as they prevent $SO_4^{2-}$ regeneration and lead to the production of $S^0$ or FeS instead (Findlay et al., 2014; Avetisyan et al., 2021), leaving a specific fingerprint in the TA and DIC effluxes.

While TA controls the $CO_2$ buffering capacity of the ocean, riverine input of carbonate to the ocean is poorly constrained (Middelburg et al., 2020) and only rare publications take into account the estuarine transformations of the carbonate species (e.g. Su et al., 2020a; Abril et al., 2003) furthermore in a context of oxygen depletion (e.g. Abril et al., 2004). To better constrain the carbonate cycle in oxygen depleted estuaries, we sampled a stratified water column in the Chesapeake Bay eleven times over two campaigns with a high vertical resolution (down to 10 cm). This protocol allows a precise description of carbonate dynamics over a redox gradient along with the measurement of the speciation of most redox sensitive elements. Such sampling illustrates carbonate chemistry on transitioning from oxygenated waters to waters devoid of oxygen as usually

only encountered in sediments or in anoxic lakes or seas (*e.g.* Black Sea). The original observed changes of alkalinity versus dissolved inorganic carbon changes are interpreted based on typical geochemical reactions occurring along the redox gradient.

## 2 Materials and methods

### 2.1 Sampling

During two sampling campaigns from August $3^{rd}$ to $9^{th}$, 2017 and July $28^{th}$ to August $3^{rd}$ 2018, eleven profile casts were
conducted in a single station in the Chesapeake Bay with a water depth of 25 m (Station 858, 38°58.54'N; 076°22.22'W). The Susquehanna River is the main tributary of the bay representing on average 2/3 of the fresh water input (Zhang et al., 2015). Despite similar season, the two campaigns occurred at very different river flow with about 850 $m^3$ $s^{-1}$ in 2017 versus 8500 $m^3$ $s^{-1}$ in 2018 due to release of flood waters from the Conowingo Dam. The August 2018 condition corresponds to flooding which occurs on average every 3.5 years (return period of 3.5 years, USGS survey).

Each CTD cast was performed during low or high tide slack periods. An oxygen sensor (Clark electrode, SBE Inc.; detection limit of 1 μM) and fluorescence sensor (Eco-FL Fluorometer, WETLabs) were part of the CTD Rosette to take measurements during sampling. Also, a submersible all plastic pump profiler was attached with the pump near the sensor orifices allowing measurement and sampling at a resolution of a few centimetres over 25 m water depth. Water was pumped to the deck within 1 minute and water passed through a flow through voltammetry system measuring continuously $O_2$, Mn(II),
Fe(II), organically complexed Fe(III), FeS clusters, $H_2S$ and polysulfides (Hudson et al., 2019). When redox interfaces were identified, samples were filtered through an acetate cartridge filter (pore size 0.45 μm) for pH and inorganic carbon parameters, which were processed onboard within a few hours after sampling in order to conserve chemical speciation. The pump profiler system was cleaned with deionised water (18 MΩ) onboard the deck of the ship after deployment. No coating effects were observed with the pump system.

**2.2 Discrete Measurements**

For each sample, all redox species were determined in the through flow voltammetry system using cyclic voltammetry with a 100 μm diameter Au/ Hg amalgam PEEK microelectrode prepared according to Luther et al. (2008) connected to a DLK-60 electrochemical analyser from Analytical Instrument Systems Inc. The detection limit of this method is 0.2 μM for sulfide and polysulfides. Discrete samples for the determination of $NO_2^-$, Fe and Mn species were filtered through nylon luer-lock syringe
filters (Millipore, 0.20 μm). Iron was measured on both bulk and filtered samples using the ferrozine method (Stookey, 1970): after HCl acidification (for Fe(II)) and an optional reduction step (for Fe(III)+Fe(II)) with hydroxylamine hydrochloric (final concentration 0.7 M for 1 hour), ammonium actetate (final concentration 0.5 M) and ferrozine (final concentration 1 mM).Absorbance at 562 nm was read with a diode array spectrophotometer (Hewlett Packard 8452B). Limit of detection is 100 nM for Fe(II) and Fe(III) with a 1-cm cell. Shipboard nitrite determination was performed using the method of Grasshoff

(1983). To 25 ml of sample, 0.5 ml of 58 mM sulfanilamide in 10% v/v HCl and 0.5 ml of a 4 mM N(1-naphthyl)ethylene diamine hydrochloride solution were added. Samples with added reagents were shaken and left to sit for 15 min, followed by UV–Vis analysis at 540 nm using a 10-cm cell to increase detection limits. Calibration curves were constructed using sodium nitrite. Limit of detection is 10 nM for $NO_2^-$.

Dissolved manganese was determined by displacement of a Cd(II)-porphyrin complex with Mn(II) to form the Mn(II)-
porphyrin complex (Ishii et al., 1982). Mn(III) species were identified based on slower reactivity with the Cd complex (Madison et al., 2011) as modified in Thibault de Chanvalon and Luther (2019). Alternatively, Mn(III) species were identified after HCl treatment (down to pH=1.5) followed by filtration in order to flocculate and eliminate the dissolved manganese bound to humic material by filtration (Oldham et al., 2017b). Limit of detection is 50 nM for Mn(II) and Mn(III) in a 1-cm cell. MnOx was measured on 20 mL samples of suspended material retained on  0.2 µm filters by the Leucoberbelin blue
(LBB) method (Jones et al., 2019). Four millilitres of a reagent solution ([LBB]= 78 µM, [acetic acid]=14mM) react with the filter and the absorbance is read at 624 nm. $KMnO_4$ was used to calibrate the LBB method which allows the calculation of the electron equivalents obtained from particulate MnOx. Results are given in as $MnO_2$ equivalent with a limit of detection of 0.1 µM and an uncertainty below 5%.

The DIC samples were preserved in 250-mL borosilicate glass bottle with 50 µL saturated $HgCl_2$ solution. The total
alkalinity (TA) samples were not poisoned to prevent HgS precipitation and $H^+$ release in anoxic and low salinity waters (Cai et al. 2017). Then TA was analysed by Gran titration in an open-cell setting (AS-ALK2, Apollo Scitech) within 24 h of collection (Cai et al. 2010a).  The DIC samples were measured by a nondispersive infrared analyzer (AS-C3, Apollo Scitech) within a week (Huang et al., 2012). The precision for DIC and TA was about 0.1%. Both DIC and TA measurements were calibrated against certified reference materials (CRMs Batch 163 and Batch 173 provided by Andrew Dickson of the Scripps
Institution of Oceanography). The pH samples were measured onboard at 25 °C within 1 h of collection using an Orion Ross glass electrode, and calibrated with NIST standard buffers. The $p$CO$_2$, calcite saturation and TA were calculated from measured DIC and pH via CO2sys program using Cai and Wang (1998) constants. The measured TA was found highly correlated to the calculated TA ($r^2$ = 0.995 and 0.998, slope = 0.995 and 1.017 for 2017 and 2018 campaign respectively), and their difference was always below 45 µM with an average of 7.5 µM for 2017 and of 22.2 µM in 2018. These results suggest low contribution
of non-carbonate species (e.g. nitrite, ammonium or organic matter (Cotovicz Jr. et al., 2016)) and measured TA was used for the interpretation.

### 2.3 Models of biogeochemical process on TA and DIC

### 2.3.1 Identification of biogeochemical processes from scatter plots

This section identifies the required conditions to interpret a scatter plot of two species in term of chemical reaction
stoichiometry: the "reaction driven" approximation. In case of turbulent diffusion mixing (sometimes called eddy diffusion)

in only one direction (no lateral input), at steady state, on a portion of space where occurs one chemical reaction, the changes of concentration, C and D, of two species can be described by equation (1) (see Appendix 1 for more details):

$$\frac{dC}{dx} = \frac{\alpha_C}{\alpha_D} \frac{dD}{dx} + G \qquad (1)$$

with $\alpha_C$ and $\alpha_D$ the respective stochiometric coefficient, Ds the effective diffusion coefficient, x the direction in the space and G an unknown constant. Here, C and D are concentrations of any elements produced or consumed during the reaction and can be TA, DIC, or a linear combination with salinity. When G is null, equation (2) is valid and the data points would be distributed along a straight line in a C versus D scatter plot.

$$\frac{dC}{dD} = \frac{\alpha_C}{\alpha_D} \qquad (2)$$

Theoretically, equation (2) is valid in at least three cases. First, if the contribution from one endmember is negligible compared to the time scale of reactions, which corresponds also to a system with only one endmember. Second, when the slope of $\Delta C/\Delta D$ between the endmembers equal $\alpha_C/\alpha_D$ which is achieved, in particular, if one endmember was previously generated from the other endmember by a chemical reaction with similar stoichiometry. And third, when the two endmembers have similar concentrations (see Appendix 2), as in this study where the excess of TA (TAex) and the excess of DIC (DICex) are calculated by linear combination with salinity to be equal to zero for the upstream and downstream endmembers. Thus, if only one reaction occurs in-between the endmembers, the slope $\Delta TAex/\Delta DICex$ would correspond to the stoichiometry of $\alpha_{TA}/\alpha_{DIC}$. The oceanic endmember for both August campaigns was the one proposed by Su et al. (2020a). The oceanic endmember varies mainly with season (Cai et al., 2020) and a maximal change of 50 µM results in 5% uncertainty on the slope of the mixing line. For the upstream estuary, larger variations exist mainly due to changes of weathering intensity and river discharge (Meybeck, 2003; Joesoef et al., 2017), and a one-off endmember has to be determined by fit with the in situ measurements at the lowest measured salinity. Between the two campaigns, the upstream endmember changed by 77 µM generating 5% of change on the slope (see Fig. A1). The upstream endmember is not a river endmember (Su et al., 2020a) but corresponds to a salinity above 1.5 avoiding any interpretation for biological activity in the fresh water part of the estuary (Meybeck et al., 1988). However, it corresponds to a larger water mass pool, less sensitive to short term changes, with a residence time being higher than 240 days in the Chesapeake upstream part (Du and Shen, 2016), and thus is more likely to satisfy the condition of steady state.

The relative uncertainty on $\Delta TAex/\Delta DICex$ is equal to the sum of the relative uncertainty of $\Delta TAex$ and $\Delta DICex$. Posing $\Delta TA$ the change of TA measured, $\Delta S$ the change of salinity and sml_TA, the slope of the mixing line for TA, we have $\Delta TAex = \Delta TA - sml\_TA \times \Delta S$. Uncertainty on $\Delta TA$ and $\Delta S$ are negligible to the relative uncertainty of the slope of the mixing line. Posing $\delta(x)$ as the uncertainty on x, we get equation (3) that describes the fact that the uncertainty is much lower on $\Delta DICex$ than on DICex because most the error associated with the calculation of the endmember is cancelled when calculating the difference of DICex on two points with close salinity:

$$\frac{\delta\left(\Delta TAex/\Delta DICex\right)}{\Delta TAex/\Delta DICex} = \frac{\delta(\Delta TAex)}{\Delta TAex} + \frac{\delta(\Delta DICex)}{\Delta DICex} = \frac{\delta(sml\_TA)}{sml\_TA} + \frac{\delta(sml\_DIC)}{sml\_DIC} = 0.1 \qquad (3)$$

However, in a stratified water column, not only one but several successive reactions occur, limiting the validity of equation (1) to each reactional stratum. The general case is not straightforward to solve, but in the particular case where the C versus D plot represents a straight line between two endmembers with different concentrations, the previous analyse of equation (2) indicates (second case) that one endmember would have been previously generated from the second by a chemical reaction with similar stoichiometry. Thus, the depths corresponding to the straight line define a reactional stratum characterised by a

constant $\alpha_C/\alpha_D$ and delimitated by two local endmembers maintained in steady state by chemical reactions with similar stoichiometry than the one that produced them, $i.e.$ $\Delta C/\Delta D = \alpha_C/\alpha_D$. The local endmembers should have been produced before the steady state achievement, by a reaction of similar stoichiometry but the reaction could have been faster than the observed one or could have occurred in a different place, including in the sediment.

      Therefore, in a system defined between only two endmembers, away from atmospheric exchanges, in case of turbulent

diffusion mixing, at steady-state and with negligible lateral mixing, the "reaction driven" approximation allows us to interpret linear variations of TAex versus DICex as a sum of biogeochemical reactions spread all over the water column that can be broken into several reactional stratum. In each stratum, if the local $\Delta TAex/\Delta DICex$ ratio is constant, it corresponds to the apparent stoichiometry of a combination of the biogeochemical reactions occurring in this stratum. In the case of multiple simultaneous reactions in the same stratum, by posing $\alpha_C^i$ the stoichiometry of the i$^{th}$ reaction concerning the reagent C and $v^i$

the reaction rate of the i$^{th}$ reaction, we obtain:

$$\alpha_C v = \sum_i \alpha_C^i v^i$$

To maintain a global reaction rate independent to the species we have

$$v = \sum_i v^i$$

so
$$\alpha_C = \frac{1}{v}\sum_i \alpha_C^i v^i \qquad (4)$$

      Equation (4) indicates that the apparent stoichiometry in a given stratum corresponds to the sum of the stoichiometric coefficients of each reaction weighted by the relative rate of each reaction. Therefore, to estimate the relative rate of each reaction to the observed local changes of TAex, DICex and AOU or H$_2$S, a linear combination of reactions is calculated. This

combination has to fit 3 equations (one for each parameter), which allows a maximum of 3 reactions to be used to solve the system. A limited number of reactions is selected as candidates based on the discussion (see Table 1 and sections 4.2 and 4.3). Then, the system is solved with the minimum possible reactions, and the weighted coefficients, $v^i/v$, are calculated.

### 2.3.2 TA changes indicated by the charge transfer during a reaction

The simplest way to calculate the TA changes induced by an individual reaction is to look at charge transfer induced by the stoichiometry of the given reaction. Indeed, the total alkalinity (TA) corresponds to the quantity of acid added to titrate a solution down to pH 4.5 (Dickson, 1981). It can be described by equation (5), with the example of HCl as acid and $B^-$ any titrated base.

$$HCl + B^- \rightarrow BH + Cl^- \tag{5}$$

Assuming a complete reaction, the quantity of acid added is equal to the negative charges initially present in the sample consumed plus the positive charges added to the species initially present in the sample. Thus, the total alkalinity corresponds to the loss of negative charges (or gain of positive charges) for species initially present in the sample produced by the pH change from the initial pH = pHini to pH = 4.5. Writing $z_i^{pH}$, the charges held by species $i$, initially present in the sample at a given pH, we get equation (6):

$$TA = \sum_i z_i^{pH=4.5} - \sum_i z_i^{pHini} \tag{6}$$

However, the electroneutrality of water induces:

$$\sum_i z_i^{pHini} = 0$$

so
$$TA = \sum_i z_i^{pH=4.5} \tag{7}$$

Equation (7) demonstrates than the total alkalinity is simply the sum of charges that each species present in the sample would have at pH 4.5. From Eq. (7), one can easily deduce the changes of alkalinity from any reaction stoichiometry as soon as the charges of all species at pH = 4.5 are known. For example, for a 0.0020 M NaOH dissolution, its TA is 0.0020 M as at pH=4.5, TA= ([$Na^+$] + [$H^+$] - [$OH^-$]) = 0.002 + $10^{-4.5}$ − $10^{-14+4.5}$ = 0.0020 M. For natural waters, most of the time, the only acid/base species charged at pH = 4.5, are $H_2PO_4^-$ and $NH_4^+$. In that respect, whatever the initial pH and the acid-base equilibrium of species in the sample is, the sum of phosphate species will count negatively and the sum of ammonium species will count positively. Strictly speaking, at pH=4.5, acid species with pKa between 2.5 and 6.5, such as $F^-$ and $NO_2^-$, would be only partially titrated and the charge equals their concentration multiplied by a correction factor of $(1+10^{pKa-4.5})^{-1}$, but this correction can be neglected to a first approximation. Eq. (7) is equivalent to those published in Soetaert et al. (2007) or Wolf-Gladrow et al. (2007) whose equation 32 can be refined considering that :

$$\sum_i z_i^{pH=4.5} = [Na^+] + 2\,[Mg^{2+}] + 2\,[Ca^{2+}] + [K^+] + 2[Sr^{2+}] + \cdots - [Cl^-] - [Br^-]$$

$$- [NO_3^-] - \cdots TPO4 + TNH3 - 2TSO4 - THF - THNO2 - \cdots$$

However, Eq. (7) is more concise and more general. For example, in suboxic water, specific species such as polysulfides (as $HS_8^{2-}$, Rickard and Luther, 2007) and in highly productive environments, carboxylic groups from DOC can be easily added as soon as the bearing charges at pH = 4.5 are known.

## 3 Results

### 3.1 Water column stratification

High-resolution profiles of salinity and temperature plotted in Fig. 1 show the stratified water column at station 858. Carbonate and redox chemistry are plotted against salinity in Fig. 2. These data result from the 11 CTD casts performed over 1 week during each campaign and correspond to depths ranging from 0.7 to 25 meters depth. Plots against depth generate noisier profiles are shown in Fig. A2 while plots against salinity follow the water masses. Despite the overall lower salinity due to a near 7-fold river flow increase in 2018 than 2017 (5800 m s$^{-1}$ in 2018 versus 850 m$^3$ s$^{-1}$ in 2017), similar zonation of the water column occurred.

A surface layer sampled only in 2018 is visible in the top 3 m depth (Fig. 1). It presents highly variable temperature and oxygen concentration and oversaturation of pCO2 (Fig. 2) indicating export to the atmosphere. Below, at 3 m depth, a subsurface layer (named primary production zone or PP in Fig. 2) is characterized by a high amount of $O_2$ (about or above 100% saturation), high pH (about 8; 8.11 ± 0.07, n=13 in 2017 and 7.94 ± 0.08, n=14 in 2018) and high day to day temperature variation (above 1 °C between different days). The layer presents relatively low pCO$_2$ (505 ± 75 µatm, n=13 in 2017 and 770

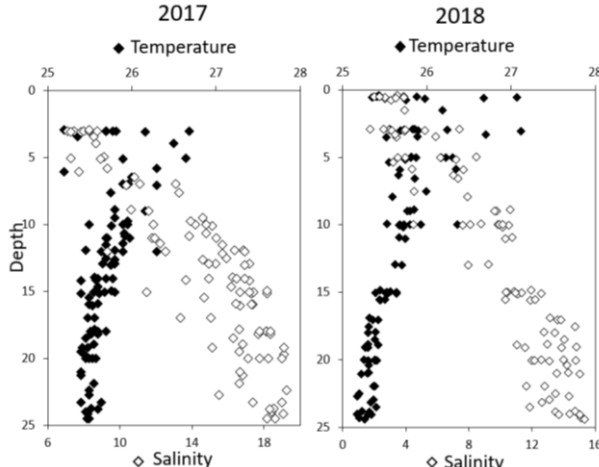

**Figure 1: Superimposed salinity (in white) and temperature (in black) profiles over the 11 casts for each campaign.**

± 130 µatm, n=14 in 2018) with minimal values at 110 µatm in 2017 and 205 µatm in 2018, which are below the atmospheric pCO₂ of 407 µatm (Chen et al., 2020)). This signature corresponds to primary production (PP). Fluorescence (not shown) correlates with pH as expected for primary production (pH = Fluo (mV) x 13 +7.14, $r^2$=0.8 in 2017 and $r^2$=0.9 in 2018). DICex and TAex reach their minimal value in the surface layer. While DICex minimum is similar between the two campaigns (78 ± 17 µM, n= 13 in 2017 and 76 ± 10 µM, n=14 in 2018); TAex minimum is much lower in 2017 (7 ± 3 µM, n=12) than in 2018

(48 ± 7 µM, n=14).

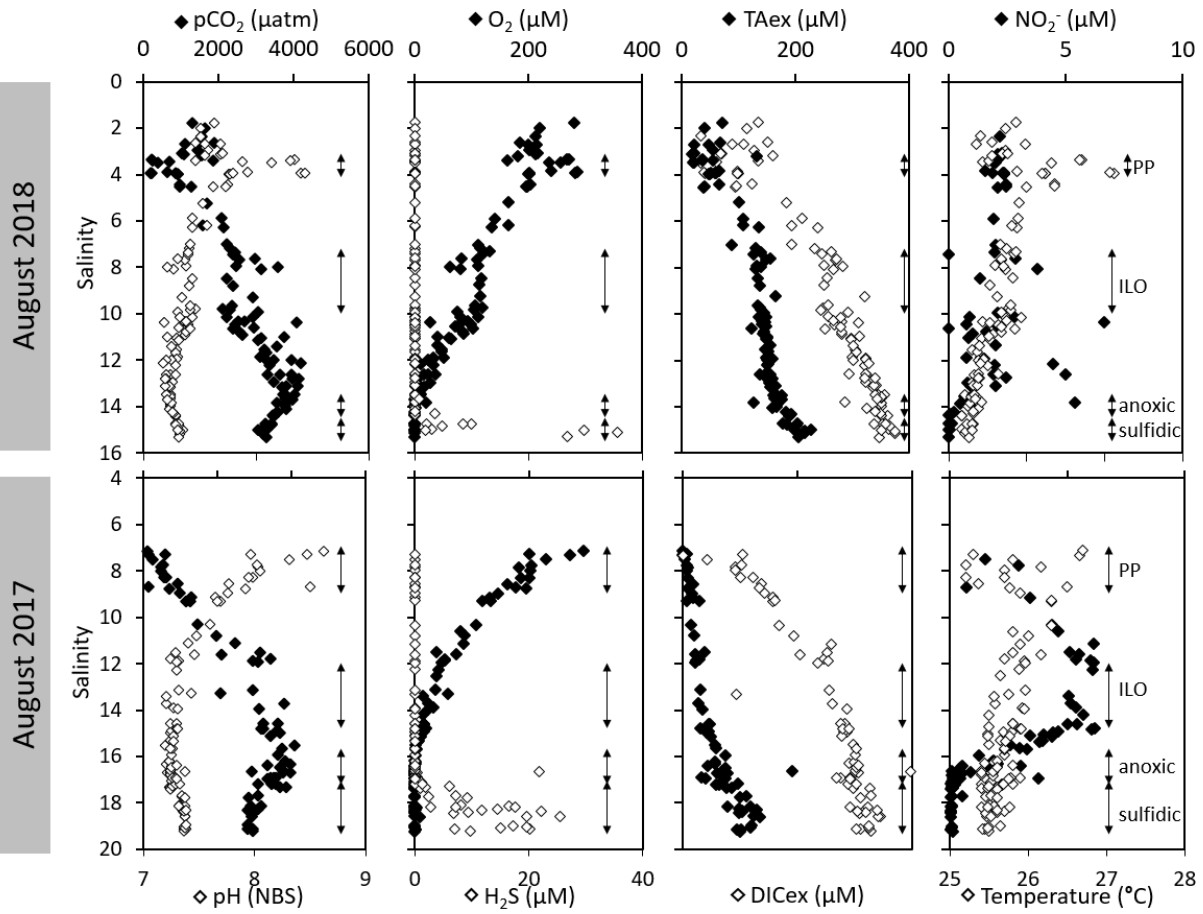

**Figure 2: Superimposed carbonate and redox chemistry profiles of 11 casts done in August 2017 and in August 2018 at the station 858**

Below, with increasing salinity, an important increase of pCO₂ accompanying the decrease of O₂, pH and temperature is visible. A relatively invariable low oxygen zone (called ILO in Fig. 2) is here defined by the salinity invariance of O₂ concentrations at about 24 ± 3 µM (n=15) in 2017 and 105 ± 5 µM (n =16) in 2018. Other species are also relatively stable for this depth such as pCO₂, at about 3250 ± 150 µatm (n=10) in 2017 and 2590 ± 100 µatm (n = 16) in 2018, and pH, about 7.28

± 0.01 (n=13) in 2017 and 7.37 ± 0.02 (n=16) in 2018. Deeper, where the oxygen is not detectable (< ~ 1 µM) and in absence of free sulphide, the so-called anoxic zone corresponds to a pH minimum at 7.24 ± 0.01 (n=40), similar in 2017 and 2018 that

generates a pCO$_2$ maximum. The deepest layer is a sulfidic layer, [H$_2$S] = 11.2 ± 2.8 µM (n=36), in which the pH seems quite stable at 7.33 ± 0.01 (n=29) and 7.32 ± 0.01 (n=9) in 2017 and 2018 respectively. The Ca$^{2+}$ concentrations observed by Su et al. (2021) and during the 2018 cruise (data not shown) vary linearly with salinity (calcium excess stays below 200 µM or 10% of total Ca). Assuming similar behaviour in 2017, calculations show that the whole water column (except 4 samples from the PP zone) is under saturated (0.36<Ωcal<1; mean=0.68) with respect to calcite in 2018, while undersaturation is only valid below S=10 in 2017. The main changes between the two campaigns correspond to a greater oxygen penetration in 2018, especially visible in the ILO zone. Additionally, a surface layer (with salinity below 3) is visible above the primary production zone in 2018 but is related to the more superficial sampling in 2018.

### 3.2 Intermediate redox species

The development of the anoxic zone during summer (Su et al., 2021) and the regularity of this development over the years (Sholkovitz et al., 1992; Trouwborst et al., 2006; Lewis et al., 2007; Cai et al., 2017; Oldham et al., 2017a), requires the presence of species able to rapidly oxidise the H$_2$S mixing upward and to reduce the O$_2$ mixing downward. The three main redox couples known to play this role, NO$_3^-$/NO$_2^-$, MnO$_x$/Mn$^{2+}$ and Fe$^{3+}$/Fe$^{2+}$ are described in Fig. 3 by the superimposition of all cast results against salinity. Four representative casts are plotted in the Fig. A3. The primary production zone is depleted

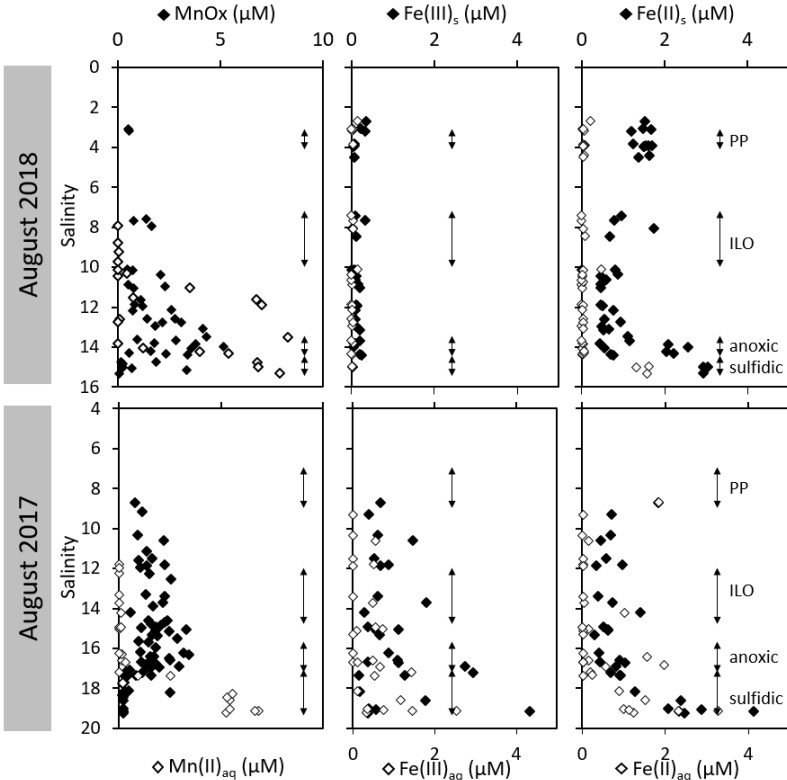

**Figure 3: Mn, Fe and nitrite profiles build by superimposition of 11 casts for each campaign.**

in dissolved Mn(II) and Mn(III), and shows an average value for solid MnOx ($0.7 \pm 0.1 \mu M$, n=4) and $NO_2^-$ ($2.1 \pm 0.1 \mu M$, n=18). In 2018, iron speciation was investigated in the primary production zone showing an important iron pool ($1.6 \pm 0.1$ $\mu M$, n=9) dominated by solid Fe(II) ($95 \pm 2\%$). Below, in the ILO zone, $NO_2^-$ reaches a maximum plateau at $5.3 \pm 0.2 \mu M$, n=11 in 2017 that is not visible in 2018, MnOx dominates the Mn pool in the ILO zone with concentration of $1.8 \pm 0.1 \mu M$ , n=14 in 2017 and $1.3 \pm 0.2 \mu M$, n=3 in 2018). Fe(II) represents only 49% $\pm$ 13, n=4 of the iron pool (for a total iron concentration of $2.0 \pm 0.4 \mu M$) in 2017 while it represents $87 \pm 5\%$, n=4 of the iron pool in 2018 (for a total concentration of $1.2 \mu M \pm 0.2$, n=4). Just below oxygen depletion, in the anoxic zone, MnOx reaches a maximum ($2.0 \pm 0.1 \mu M$, n=25 in 2017 and $2.9 \pm 0.4 \mu M$, n=12 in 2018). During both campaigns, the sulfidic layer is characterised by the absence of $NO_2^-$ and MnOx while dissolved manganese concentration increases up to $7.7 \pm 0.8 \mu M$, n=15 and the iron pool increases to $5.0 \pm 0.7$ $\mu M$, n=13. No Mn(III) was detected with the porphyrin kinetics method (Thibault de Chanvalon and Luther, 2019) but about 30% of the total dissolved manganese flocculated after acidification down to pH 1.5 when analysed in 2017 indicating the existence of Mn(III).

## 4. Discussion

In terms of salinity, Station 858 of the Chesapeake Bay shows very similar water column zonation between summer 2017 and summer 2018 despite a 10-fold difference in freshwater discharge rates (Fig. 1 and Fig. 2). Major features are: first, a surface layer characterized by intense atmospheric exchange that was only sampled in 2018. Below, at about 3 meter depth, a subsurface layer associated with high primary production (PP zone) with high pH about 8, oversaturation of dioxygen and low $CO_2$ partial pressure (down to 110 µatm). Below, a low oxygen layer with invariant concentration of most species surveyed (the invariant low oxygen (ILO) zone) is characterized by significant nitrite accumulation in 2017 (Fig. 2), probably due to oxidation of $NH_4^+$ diffusing upward and/or produced by *in situ* remineralisation. This feature is not visible in 2018 probably because the higher $O_2$ concentration in 2018 accelerates nitrite oxidation into nitrate and prevents any significant accumulation. Below, the anoxic zone, with neither $O_2$ nor $H_2S$ detectable, is characterized by an increase of MnOx concentration and a pH minimum. This MnOx maximum can be explained by the upward diffusing $Mn^{2+}$ that is biologically oxidized by the downward diffusing $O_2$, though at low, undetectable, concentration (Clement et al., 2009). Additionally, $Mn^{2+}$ could be oxidized by the nitrite or the nitrate (not measured) diffusing downward (thermodynamically favourable (Luther, 2010)). Compared to the ILO zone, the anoxic MnOx maximum corresponds to an increase of $0.2 \mu M$ in 2017, while it is much more marked in 2018 with an increase of $1.6 \mu M$ (Fig. 3). This difference could come from a faster $Mn^{2+}$ oxidation produced by the steeper oxygen gradient above the anoxic zone and by the thinness of the anoxic layer in 2018. Finally, in the deeper sulfidic layer, the MnOx disappearance corresponds to the $Mn^{2+}$ increases (Fig. 3) according to the reduction of settling MnOx by $H_2S$. The concentration increase of the manganese pool and of the iron pool with depth in the anoxic and sulfidic layers probably results from important sedimentary efflux.

### 4.1 Validity of the "reaction driven" approximation

The simultaneous and high-resolution sampling of multiple carbonate parameters and redox species gives us the rare opportunity to investigate in detail the interaction between carbonate species and redox sensitive elements. In the Chesapeake Bay, main changes of DIC and TA can be explained by mixing between upstream and oceanic endmembers (Appendix, Fig. A1). This "endmember driven" interpretation leads to the calculation of an excess of DIC and TA, DICex and TAex, relative to the mixing line as shown in Fig. 4a. At station 858, the steep gradient observed, for example the pH and $pCO_2$ gradients in the PP zone, the $O_2$ and $NO_2^-$ gradients above the anoxic zone and the Mn, Fe and $H_2S$ gradients at depth, suggest that ongoing in situ processes control the changes of concentrations and dominate the time-dependent endmember variability or the mixing with an unknown third endmember. Additionally, the TAex versus DICex plot (Fig. 4a) shows steep changes of direction, such as in case of straight lines between local endmembers maintained by ongoing reactions while a preponderance of mixing would produce more progressive changes. Finally, at depth, Fig. 4a shows a similar slope for both years studied ($\Delta TAex/\Delta DICex$ = 2.4) rather than a similar TAex and DICex concentration, which reinforces the validity of the "reaction driven" approximation.

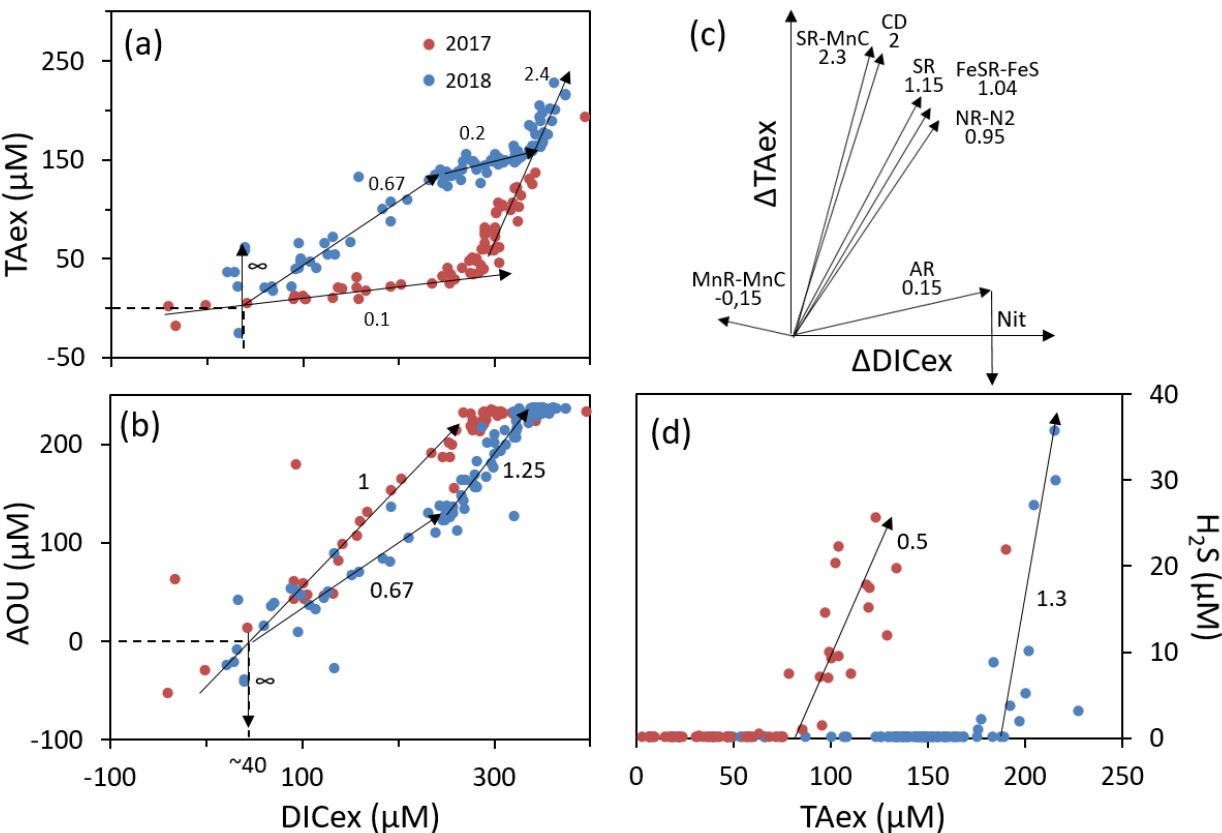

**Figure 4: Description of the TAex/DICex/AOU or $H_2S$ system. (a), (b) and (d) panels show samples measured along with interpretative slopes discussed in the main text. Panel (c) shows the theoretical slope of TAex/DICex from combination of reactions presented in Table 1.**

Assuming 1) that mixing is efficiently described by vertical turbulent diffusion mixing, 2) that the measured concentrations correspond to a steady state – no changes observed over the 1 week sampling, 3) that no additional endmember contributes significantly to the excess calculation, in particular that the samples are isolated from atmospheric exchanges and 4) that lateral mixing is negligible, which is equivalent to the lateral invariance of the system – as in the stratified water column of station 858; the "reaction driven" approximation (section 2.3.1) permits interpretation of the concentration changes as a linear combination of the stoichiometry of several chemical reactions (equation 5). The weighted coefficients of each reaction are equal to the rate of each reaction relative to the sum of the rate of all occurring reactions, $v^i / v$ (equation 7). Accordingly, this interpretation does not identify reactions with minor impact on the carbonate cycle (equation 7) or reactions that cancelled each other later during the journey. For example, $\Delta TAex$ introduced by PP is frequently cancelled by a similar amount or even

**Table 1: Summary of the main net reactions occurring among the different zones of a redox gradient with reactants starting in equilibrium with the atmosphere, adapted from Soetaert et al. (2007). The calculations assume Redfield ratio of the organic matter, i. e. $\gamma N = 0.156$ and $\gamma P = 0.0094$. Redox zones describe the different conditions required for completing the reaction.**

| Name | Redox zones | Net Formula | $\Delta TA$ | $\Delta DIC$ |
|---|---|---|---|---|
| AR | Oxic | $(CH_2O)(NH_3)_{\gamma N}(H_3PO_4)_{\gamma P} + O_2 \rightarrow$ $\gamma^N\,NH_3 + \gamma^P\,H_3PO_4 + CO_2 + H_2O$ | 0.15 | +1 |
| CD | Any | $CaCO_3 + CO_2 \rightarrow Ca^{2+} + 2\,HCO_3^-$ | 2 | +1 |
| Nit | Oxic | $NH_3 + 2\,O_2 \rightarrow HNO_3 + H_2O$ | -2 | 0 |
| NR-N2 | Anoxic | $(CH_2O)(NH_3)_{\gamma N}(H_3PO_4)_{\gamma P} + 0.8\,HNO_3 \rightarrow$ $\gamma^N\,NH_3 + \gamma^P\,H_3PO_4 + CO_2 + 0.4\,N_2 + 1.4\,H_2O$ | 0.95 | +1 |
| NR-NH3 | Anoxic | $(CH_2O)(NH_3)_{\gamma N}(H_3PO_4)_{\gamma P} + 0.5\,HNO_3 \rightarrow$ $\gamma^N\,NH_3 + \gamma^P\,H_3PO_4 + CO_2 + 0.5\,NH_3 + 0.5\,H_2O$ | 1.15 | +1 |
| MnR-MnC | Anoxic | $(CH_2O)(NH_3)_{\gamma N}(H_3PO_4)_{\gamma P} + 2\,MnO_2 + CO_2 \rightarrow$ $\gamma^N\,NH_3 + \gamma^P\,H_3PO_4 + 2\,MnCO_3 + H_2O$ | 0.15 | -1 |
| SR | Sulfidic | $(CH_2O)(NH_3)_{\gamma N}(H_3PO_4)_{\gamma P} + 0.5\,H_2SO_4 \rightarrow$ $\gamma^N\,NH_3 + \gamma^P\,H_3PO_4 + CO_2 + 0.5\,H_2S + H_2O$ | 1.15 | +1 |
| SR-SMnC | Sulfidic | $(CH_2O)(NH_3)_{\gamma N}(H_3PO_4)_{\gamma P} + 1/2\,H_2SO_4 + 0.5\,MnO_2 \rightarrow$ $\gamma^N\,NH_3 + \gamma^P\,H_3PO_4 + 1/2\,CO_2 + 1/2\,MnCO_3 + 0.5\,S^0 + 1.5\,H_2O$ | 1.15 | +0.5 |
| FeSR-FeS | Sulfidic | $(CH_2O)(NH_3)_{\gamma N}(H_3PO_4)_{\gamma P} + 0.44\,FeOOH + 0.44\,H_2SO_4 \rightarrow$ $\gamma^N\,NH_3 + \gamma^P\,H_3PO_4 + CO_2 + 0.44\,FeS + 1.67\,H_2O$ | 1.04 | +1 |
| SR-SFeS | Sulfidic | $(CH_2O)(NH_3)_{\gamma N}(H_3PO_4)_{\gamma P} + 0.33\,FeOOH + 0.5\,H_2SO_4 \rightarrow$ $\gamma^N\,NH_3 + \gamma^P\,H_3PO_4 + CO_2 + 0.33\,FeS + 0.17\,S^0 + 1.67\,H_2O$ | 1.15 | +1 |
| SR-FeS2 | Sulfidic | $(CH_2O)(NH_3)_{\gamma N}(H_3PO_4)_{\gamma P} + 0.27\,FeOOH + 0.53\,H_2SO_4 \rightarrow$ $\gamma^N\,NH_3 + \gamma^P\,H_3PO_4 + CO_2 + 0.27\,FeS_2 + 1.67\,H_2O$ | 1.21 | +1 |

higher ΔTAex by AR. The starting point corresponds to the upstream (S=1.5) endmember at DICex=0; TAex=0. Figure 4a shows a slight DICex enrichment at TAex=0 which reflects a bias from the endmember calculation.

**4.2 Identification of major reactions in oxygenated water column using ΔTAex/ΔDICex/ΔAOU signature.**

In oxygenated water, the ΔTAex/ΔDICex analysis can be supported and strengthened by looking also at the ΔAOU/ΔDICex ratio (Fig. 4b) and establishing a ΔTAex/ΔDICex/ΔAOU signature for each water layer. Then, a linear combination of up to 3 reactions can be fit to the observed ΔTAex/ΔDICex/ΔAOU signature. The main candidate reactions to be combined are: aerobic respiration (AR), primary production (whose overall mass balance equation is here summarized as negative AR), carbonate dissolution (CD) and nitrification of ammonium (Nit) as presented in Table 1 and in Fig. 4c. In 2017, the whole oxygenated zone, including the PP zone and the ILO zone, is characterised by a ΔTAex/ΔDICex/ΔAOU = 0.1/1/1 (Fig. 4a and 4b) which corresponds to the occurrence of only net aerobic respiration (AR) (theoretical values are 0.15/1/1 see Table 1 and Fig. 4c). Note a - that "net aerobic respiration" indicates that primary production is possible at a significant rate, but slower than AR; and b – that this ΔTAex/ΔDICex/ΔAOU signature indicates weak nitrification following respiration. In case of full nitrification, the theoretical slopes should be ΔTAex/ΔDICex/ΔAOU = -0.167/1/1.31 as proposed by Zeebe and Wolf-Gladrow (2001). A combination of 1 AR and 0.025 Nit (nitrification of 16% of the produced $NH_3$) improves the modelled value to 0.1/1/1.05. The relatively slow nitrification can be explained by slow kinetics of $NH_4^+$ oxidation, with a half-life time estimated between a few days in estuaries (Horrigan et al., 1990) to multiple years in coastal environments (Heiss and Fulweiler, 2016). As a comparison, other $NH_4^+$ fates, such as adsorption leads to an ammonium half life time of about a few minutes (Alshameri et al., 2018) to a few hours (Raaphorst and Malschaert, 1996) depending on the concentration of fine particles. Additionally, algae are known to use $NH_4^+$ as a N source (Raven et al., 1992) and $NH_4^+$ can be directly assimilated by heterotrophic organisms.

In 2018, fresh water masses brought by the exceptional flood modified the carbonate system equilibrium. First, a low salinity layer with pCO2 at 1540 µatm overlays the primary production layer (Fig. 2), preventing the uptake of atmospheric $CO_2$ by primary production as was observed in 2016 (Chen et al., 2020). Just below the surface layer, in the PP zone, the lock down of atmospheric exchange by the low salinity layer produces supersaturation of trapped $O_2$ (Fig. 2). In Fig. 4a and 4b, this process translates into a vertical distribution at DICex = 40 µM with ΔTAex/ΔDICex/ΔAOU = 1.37/0/-1. This original signature can be modelled by the combination of simultaneous carbonate dissolution (CD), the water column being undersaturated, and PP, no important turbidity was visible as modelled by Cerco et al. (2013), in equal proportion (2nd line in Table 2); the carbonate dissolution buffers the DIC consumption by the PP. Note that the ratio between ΔTAex/ΔAOU implicates an important nitrate assimilation superior or equal to the amount of N required for the PP, as modelized by negative nitrification in Table 2.

In 2018, below the PP zone down to DICex = 240µM, the beginning of the ILO zone, the TAex increases significantly with a ΔTAex/ΔDICex/ΔAOU = 0.67/1/0.67 (Fig. 4a and 4b), incompatible with AR. To explain this signature a contribution of CD superimposed on AR seems most likely. A linear combination fitting leads 0.5 CD for 1 AR (3rd line in Table 2) and

results in $\Delta TAex/\Delta DICex/\Delta AOU = 0.77/1/0.67$. An explanation of its occurrence solely in 2018 could be the increase in carbonate rich suspended material at high flow conditions (Su et al., 2021). Excess of $Ca^{2+}$ compared to the mixing line with oceanic end member (not shown) indicates that up to 200 µM of $Ca^{2+}$ is produced in the oxygenated layer. Deeper, in the ILO zone, $\Delta TAex/\Delta DICex/\Delta AOU = 0.2/1/1.25$ (Fig. 4a and 4b), which results mainly from AR (0.15/1/1) with possible addition of CD and Nit, the exact signature being fitted for 0.54 CD and 0.46 Nit for 1 AR (4th line in Table 2), as a continuum of AR and CD relative rates from the overlaying layer. This important nitrification is also in good agreement with the nitrification observed in the PP zone during this campaign, the lack of nitrite build up in the ILO zone and the relatively high oxygen concentration (at 105 µM) in the ILO zone in 2018 which is able to sustain nitrification. Overall, our results show that the higher river flow of 2018 increases carbonate dissolution for the top 5-10 m water depth that is superimposed on primary production or aerobic respiration. Additionally, in 2017 the $\Delta TAex/\Delta DICex/\Delta AOU$ system indicates weak nitrification, while in 2018 significant nitrification in the ILO and PP zones are suggested by the "reaction driven" approximation. The role of nitrification in explaining TAex depletion is only hypothetical since no direct measurement of $NH_4^+$ and $NO_3^-$ were performed. In particular, TAex depletion is particularly intense during high flow, high suspended particles season and could be produced by $NH_4^+$ adsorption to the particles rather than by nitrification.

**4.3 Identification of major reactions in the anoxic water column using $\Delta TAex/\Delta DICex/\Delta H_2S$ signature.**

In the absence of oxygen, the $\Delta TAex/\Delta DICex/\Delta H_2S$ will be used for rates calculation. In the anoxic zone, the signature is similar in summer 2017 and 2018 at $\Delta TAex/\Delta DICex/\Delta H_2S = 2.4/1/0$. The "reaction driven" approximation of this signature is more difficult than in oxidized water because more reactions are known to occur simultaneously in absence of oxygen. However, to fit with the "reaction driven" approximation, it is not necessary to describe each reaction step but only the overall changes concerning the journey of a water mass over different redox conditions resulting in a combined result or reaction. This approach has recently been proposed for FeS burial by Hiscock and Millero (2006), Rassmann et al. (2020) or Su et al. (2020b). A scenario combining sulfate reduction (SR) is particularly attractive since SR represents the main carbon remineralisation pathway in absence of oxygen. However, a combination of SR with CD would result in a $\Delta TAex/\Delta DICex$ between 1.15 and 2 (see Table 1 or Fig. 4c) and fails to reach the $\Delta TAex/\Delta DICex$ of 2.4. Moreover, SR alone underestimates the importance of the $H_2S$ oxidation pathway that can consume all the alkalinity produced during SR. For example, SR follow by oxygenated oxidation results in $\Delta TAex/\Delta DICex/\Delta AOU$ signature equal to AR only. In the Chesapeake Bay, $H_2S$ oxidation is critical since no $H_2S$ is measurable in the anoxic zone while the gradient at the sediment/water interface indicates high $H_2S$ sedimentary efflux (Fig. 2).

Generalizing these observations, recent efforts to build an alkalinity budget on the global scale (Hu and Cai, 2011; Middelburg et al., 2020) highlight that the alkalinity produced by anaerobic respiration corresponds to the uncharged species produced, mostly in solid or gaseous phases. Indeed, the alkalinity changes produced during a natural reaction equal the "charge transfer" from species having some charge at pH = 4.5, such as $NO_3^-$ and $SO_4^{2-}$, to species that would lose its charges at pH = 4.5, mainly $HCO_3^-$, that is not counted in the alkalinity calculation (see Eq. (12)). Although correct, this approach tends to

neglect the roles of Fe and Mn oxides (Middelburg et al., 2020) since their transformation from (oxyhydr)oxides into sulphur or carbonate solid species does not involve any charge transfer. When looking in detail at these processes, the metal oxides are critical since they are the main $H_2S$ oxidation pathway that does not regenerate $H_2SO_4$ but rather produces $S^0$, $S_n^{2-}$ or FeS (Findlay et al., 2014; Avetisyan et al., 2021) which limits alkalinity consumption.

To build a pool of candidate reactions for the fitting, first, dissolved species at too low a concentration (*e.g.* $Mn^{2+}_{aq}$, $Fe^{2+}_{aq}$) to be a net reagent to affect the carbon cycle at steady state are not taken into account. These species are usually recycled rapidly and hold a role of catalyser or electron shuttle between other redox species and did not reach 10 µM during the

**Table 2: Linear combination of reactions from Table 1 that fit the observations (see text for details, H₂O molecules are omitted).**

| | | Linear combination (for 1 CH₂O) | ΔTAex / ΔDICex | ΔAOU/ΔDICex or ΔH₂S / ΔDICex | Net Formula (for 1 CH2O) |
|---|---|---|---|---|---|
| **Observed in 2018** | **Oxic** | AR+0.025Nit | 0.1 | 1.05 | $(CH2O)(NH3)\gamma N(H3PO4)\gamma P + O2 \rightarrow$ $0.16\ \gamma N\ HNO3 + 0.84\ \gamma N\ NH3 + \gamma P\ H3PO4 + CO2$ |
| | | CD-AR-0.62Nit | ∞ | -∞ | $(4\gamma^N)\ HNO_3 + \gamma^P\ H_3PO_4 + CaCO_3 + 2\ CO_2 \rightarrow$ $(CH2O)(NH_3)_{\gamma N}(H_3PO_4)_{\gamma P} + 2.24\ O_2 + Ca^{2+} + 2\ HCO_3^- + 0.46\ NH_3$ |
| | | AR+0.5CD | 0.77 | 0.67 | $(CH2O)(NH_3)_{\gamma N}(H_3PO_4)_{\gamma P} + O_2 + 0.5\ CaCO_3 \rightarrow$ $\gamma^N\ NH_3 + \gamma^P\ H_3PO_4 + 0.5\ CO_2 + 0.5\ Ca^{2+} + HCO_3^-$ |
| | | AR + 0.54 CD + 0.46 Nit | 0.2 | 1.25 | $(CH2O)(NH_3)_{\gamma N}(H_3PO_4)_{\gamma P} + 1.93\ O_2 + 0.54\ CaCO_3 + 2\ \gamma_N\ NH_3 \rightarrow$ $3\gamma^N\ HNO_3 + \gamma^P\ H_3PO_4 + 0.46\ CO_2 + 0.54\ Ca^{2+} + 1.1\ HCO_3^-$ |
| **Observed in 2017** | **Anoxic** | 0.98 SR-SMnC + 0.02 MnR-MnC | 2.4 | 0 | $(CH2O)(NH_3)_{\gamma N}(H_3PO_4)_{\gamma P} + 0.49\ H_2SO_4 + 0.53\ MnO_2 \rightarrow$ $\gamma^N\ NH_3 + \gamma^P\ H_3PO_4 + 0.47\ CO_2 + 0.53\ MnCO_3 + 0.49\ S^0$ |
| | | 0.65 SR-FeS + 0.35 MnR-MnC | 2.4 | 0 | $(CH2O)(NH_3)_{\gamma N}(H_3PO_4)_{\gamma P} + 0.3\ H_2SO_4 + 0.7\ MnO_2 + 0.3\ FeOOH \rightarrow$ $\gamma^N\ NH_3 + \gamma^P\ H_3PO_4 + 0.3\ CO_2 + 0.7\ MnCO_3 + 0.29\ FeS$ |
| | | 6.4 CD + MnR-MnC | 2.4 | 0 | $(CH2O)(NH_3)_{\gamma N}(H_3PO_4)_{\gamma P} + 6.4\ CaCO_3 + 2\ MnO_2 + 7.4\ CO_2 \rightarrow$ $\gamma^N\ NH_3 + \gamma^P\ H_3PO_4 + 12.8\ HCO_3^- + 2\ MnCO_3 + 6.4\ Ca^{2+}$ |
| | **Sulfidic** | 0.38 MnR-MnC + 0.76 SR − 0.15 SR-SMnC | 2.4 | 1.2 | $(CH2O)(NH_3)_{\gamma N}(H_3PO_4)_{\gamma P} + 0.31\ H_2SO_4 + 0.68\ MnO_2 + 0.07\ S^0 \rightarrow$ $\gamma^N\ NH_3 + \gamma^P\ H_3PO_4 + 0.32\ CO_2 + 0.68\ MnCO_3 + 0.38\ H_2S$ |
| | | 0.64 MnR-MnC + 1.36 SR − SR-SMnC | 2.4 | 3.2 | $(CH2O)(NH_3)_{\gamma N}(H_3PO_4)_{\gamma P} + 0.18\ H_2SO_4 + 0.79\ MnO_2 + 0.5\ S^0 \rightarrow$ $\gamma^N\ NH_3 + \gamma^P\ H_3PO_4 + 0.21\ CO_2 + 0.79\ MnCO_3 + 0.68\ H_2S$ |

campaigns (Fig. 3). Second, many minerals are expected to be at low concentration or thermodynamically not favoured, and their associated reactions are neglected (*e.g.,* iron phosphate, ferrous or manganous oxide, metal sulphur clusters, MnS, $FeCO_3$, adsorption processes, reverse weathering). Therefore, only aqueous species with important stock concentrations (that can exceed 0.1 mM in anoxic water) are taken into account, *i. e.,* $SO_4^{2-}$, $Ca^{2+}$, $H_2S$, $NH_4^+$ together with gaseous ($N_2$, $CO_2$) and main solid phases ($FeS_2$, $FeS$, $S^0$, $MnCO_3$, $FeOOH$, $MnO_2$). Third, the combination of many carbon remineralisation reactions with re-oxidation reactions or the net result produces a net chemical equation equal to another remineralisation reaction. As an example, the chemical equation of SR followed by $H_2S$ oxidation with oxygen is equal to the equation of aerobic respiration: the proposed model confounds both pathways because the resulting chemical changes are similar.

Table 1 lists the resulting combined reactions, and the calculated $\Delta TAex/\Delta DICex$ slopes are represented in Fig. 4c. For the anoxic zone, the nitrate respiration can be associated with $N_2$ production (NR-N2) or with $NH_3$ production (NR-NH3), and the manganese oxides respiration produces carbonate precipitation (MnR-MnC). For the sulfidic zone, SR can occur alone producing a build-up of $H_2S$. However, at a certain point, $H_2S$ gets significantly oxidised either by $MnO_2$ which produces $MnCO_3$ and $S_0$ (SR-SMnC) or by FeOOH, producing FeS and $S_0$ (SR-SFeS) and ultimately $FeS_2$ (SR-FeS2). Direct respiration of FeOOH is also taken into account, but as the only final Fe product in the model is FeS or $FeS_2$, it has to be accompanied by some SR (FeSR-FeS).

Figure 4c demonstrates that the slope of $\Delta TAex/\Delta DICex = 2.4$ can be obtained for any reaction in combination with MnR-MnC. Combinations without MnR-MnC, however, lead to a negative SR whose overall equation could be interpreted as a possible small participation of anoxygenic phototrophic (purple) bacteria (Findlay et al., 2015, 2017) but are not considered further as the amount of $\Delta TAex$ involved would be tiny. Therefore, in the absence of nitrate, oxygen and $H_2S$, only a combination of MnR-MnC with SR-SMnC (producing $S^0$, 5th line in Table 2), SR-FeS (producing FeS, 6th line in Table 2) or CD (releasing $Ca^{2+}$, 7th line in Table 2) gives the particularly high $\Delta TAex/\Delta DICex$ of 2.4. $S_0$ was not measured during our campaign, but it has been previously reported in this water column (Findlay et al., 2014), and the $S_0$ produced by SR-SMnC can react with FeS to form $FeS_2$. All the three identified combinations require a critical role of $MnO_2$. Since Figure 3 does not indicate any clear reaction for iron while the steep gradient of MnOx in proximity to the $H_2S$ rich layer suggests reaction between MnOx and $H_2S$; the combination with SR-SMnC is the most likely (5th line in Table 2). Deeper, the vertical gradient of sulphide suggests that part of the $H_2S$ came by diffusion from the sediment's porewater (Fig. 2). The assumptions required for the "reaction driven" approximation are still valid as soon as steady state is maintained by ongoing reactions, even if one of the local endmembers has not been sampled since it is probably located in the sediment. In the presence of sulphide, the $\Delta TAex/\Delta DICex/\Delta H_2S$ signature is 2.4/1/1.2 in 2017 and 2.4/1/3.2 in 2018 (Fig. 4a and 4d) and can be explained by the same combination of reactions without complete oxidation of $H_2S$ from SR to take into account the build-up of $H_2S$ (Table 2).

### 4.4 Comparison with other studies

The high observed ratio of $\Delta TAex/\Delta DICex = 2.4$ seems very specific to the Chesapeake Bay. Moreover, the "reaction driven" interpretation can be applied to other published datasets for which the $\Delta TA/\Delta DIC/\Delta H_2S$ system can be calculated

(Table 3). In the water column, most of the available datasets are not suitable for the "reaction driven" approximation since either they focus on surface water where DIC and TA are strongly impacted by atmospheric exchange or the water masses

**Table 3: Overview of the ΔTA/ΔDIC/ΔH2S signature observed in different environments**

| Publication | Sample type | a ΔTA/ΔDIC/ ΔH2S or b ΔTA/ΔDIC/ ΔAOU | ΔTA | Reaction driven interpretation |
|---|---|---|---|---|
| Hu et al., 2010 | Gulf of Mexico sediment (slope) | 1.15/1/0.53 a | + 15 mM | SR |
| Lukawska-Matuszewska, 2016 | Baltic Sea sediment | 1.3/1/0.07 a | + 13 mM | SR-FeS2 |
| Rassmann et al., 2020 | Rhone prodelta sediment | 1/1/0 a | + 5.6 mM | FeSR-FeS |
| Cai et al. 1998 | Satilla estuary | 2/1/ND | + 0.2 mM | CD |
| Abril et al. 2003 | Loire estuary | 0.88/1/0.8 | + 0.4 mM | Not applicable** |
| Drupp et al. 2016 | Oxygenated Hawai carbonate reef sands | 0.86-0.91/1/ND | + 1.5 mM | Not applicable* |
| Su et al. 2021 | Chesapeake Bay water column | 2/1/1.5 b<br>0.2/1/1 b<br>0.8/1/ND | +0.4 mM<br>+0.05 mM<br>+0.1 mM | Not applicable**<br>AR<br>NR-N2 or CD+Nit |
| Hiscock and Millero 2006 | Western Black Sea water column | 1.3/1/0.5 a | +1.2 mM | SR |

*Lack of data on oxygen concentration prevent any interpretation. ** Important air-water exchange prevents any "reaction driven" approximation.

change too fast to consider that reactions dominate over water mixing. However, porewater measured in the Gulf of Mexico has $\Delta TAex/\Delta DICex/\Delta H_2S$ = 1.15/1/0.53 (Hu et al., 2010) as expected when sulfate reduction is associated with $H_2S$ accumulation (SR reaction; $\Delta TAex/\Delta DICex/\Delta H_2S$ = 1.15/1/0.5; Table 1). Similarly, Hiscock and Millero (2006) report $\Delta TAex/\Delta DICex/\Delta H_2S$ = 1.3/1/0.5 in the Western Black Sea close to the SR signature. In the Baltic Sea sediment, $\Delta TAex/\Delta DICex/\Delta H_2S$ = 1.3/1/0.07 was reported (Lukawska-Matuszewska, 2016), which is close to the expected signature in the case of important $H_2S$ consumption by Fe oxides and consequent precipitation as pyrite (SR-FeS2 reaction; $\Delta TAex/\Delta DICex/\Delta H_2S$ = 1.2/1/0). In Rhone river prodelta sediments, the reported $\Delta TAex/\Delta DICex/\Delta H_2S$ is 1/1/0 that can be related to the 1/1/0 signature of FeSR-FeS (Table 1) that is expected in iron rich sediments with high sedimentation rates preventing $FeS_2$ formation in the pore water (Rassmann et al., 2020). In permeable, carbonate rich sediments, the reported signature of $\Delta TAex/\Delta DICex/\Delta H_2S$ in Hawaii sands is 0.86/1/ND (Drupp et al., 2016). The lack of salinity and oxygen datasets prevents further model fits, but the $\Delta TAex/\Delta DICex$ is below 2.4. For oxygen depleted data from the whole Chesapeake Bay described in Su et al. (2020), the signatures are $\Delta TAex/\Delta DICex/\Delta AOU$ = 0.2/1/1 in presence of oxygen (typical signature of AR) and 0.8/1/ND in absence of oxygen that could correspond to NR-N2 or a combination of CD+Nit. Overall, this bibliographic survey highlights the effectiveness of the "reaction driven" approximation to identify preponderant reactions controlling the carbon cycle and puts in perspective the originality of the $\Delta TAex/\Delta DICex/\Delta AOU$ signature of 2.4/1/0 observed in the Chesapeake Bay.

**4.5 Local budget**

While the "reaction driven" approximation indicates a dominant role of the SR-SMnC reaction; this possibility needs to be validated looking at the saturation state of rhodocrosite (the main $MnCO_3$ mineral) and looking at the mass budget between MnOx consumed and TAex produced. The rhodocrosite saturation (Luo and Millero, 2003) is always below 0.3 in our samples, which stands against the occurrence of *in situ* SR-SMnC reaction. When inspecting the mass budget, the 88 µM to 155 µM of $MnO_2$ required to produce the 100 µM TAex increase observed (Fig. 4a) is one order of magnitude higher than the observed MnOx or $Mn^{2+}$ concentration (Fig. 3). This mass budget discrepancy cannot be solved invoking suspended material since the 88 µM of $MnO_2$ would require a suspended material concentration of about 4.4 g $L^{-1}$ (assuming an average concentration of 20 µmol $g^{-1}$ of Mn), which is again one or two orders of magnitude higher than the 0.01 – 0.1 g $L^{-1}$ usually found in the Chesapeake Bay (Cerco et al., 2013). However, a fast-settling rate could satisfy and explain the discrepancy between water and solid concentration. But another process dephasing aqueous from solid reaction products is also possible at station 858, since the dissolved phase could have moved up, rather than the particles settling down. In this case, the SR-SMnC reaction was not happening only in the water column of the Chesapeake Bay and part of the TAex and DICex pool could have been produced in the sediment during the previous year (*e.g.* Aller, 2014), then diffused out of the sediment simultaneously with other reduced elements as the summer begins. Indeed, many previous studies at station 858 (*e.g.,* Sholkovitz et al., 1992) explained the seasonality of anoxia with an upward move of the redox front from the sediments to bottom waters during the start of summer. Important sedimentary efflux of $H_2S$, $Fe_{aq}$ and $Mn_{aq}$ were still visible during both of our August campaigns. Therefore, the 100 µM TA pool does not fit with the ambient $Mn^{2+}$ or $MnO_2$ in the water column but rather with the $MnCO_3$ deposited in the sediment. The sedimentary solid Mn stock of the Chesapeake Bay is particularly important, up to 70 µmol $g^{-1}$ at station 858 (Sinex and Helz, 1981) compared to an average value of 15 µmol $g^{-1}$ for the upper continental crust (Rudnick and Gao, 2003). Indeed semi-enclosed basins are known to concentrate manganese at the deeper sediment (Thamdrup and Dalsgaard, 2000; Lenstra et al., 2020). Recent investigations at a close station (ET 5.1; 38°48.36'N; 75°54.66'W) in the Chesapeake Bay (Lenstra et al., 2021) report that about 60% of the surface sedimentary Mn pool is MnOx (acid ascorbic extractable) and 25 % is Mn carbonate (1M HCl extraction). Assuming a porosity of 0.8, a bulk solid density of 2.6, the sedimentary pool corresponds to 35 mM of manganese which largely exceeds the 88 µM required to produce the 100 µM TAex increase. Therefore, the Chesapeake Bay sediment is particularly rich in manganese and could host important SR-SMnC reactions in the surface pore water whose soluble products diffuse up to the water column during summer and could bear with them the high ΔTAex/ΔDICex signature observed.

The assumptions detailed in section 4.1 permit the "reaction driven" approach to be reconsidered in regard to this sediment efflux scenario. Indeed, the sediment efflux does not need to be considered as an additional endmember, that would violate the third assumption, since its salinity and the pore water concentrations results from the upstream and oceanic endmembers superimposed to geochemical reactions. In section 2.3.1, we point out that equation (1) was valid in between each local endmembers and that the straight lines on the ΔTAex/ΔDICex plot between them indicates that the local endmembers are

maintained in steady state by a chemical reaction with a similar stoichiometry. These results indicate that even if most of the $MnCO_3$ was produced when the local endmembers were localised in the sediment, a- their migration does not alter the TAex/DICex signature and b- the chemical reaction that produced them is still ongoing at sufficient rate to maintain a steady state characterised by the steep changes of direction observed in Fig. 4a.

**Conclusion**

The "reaction driven" approximation proposed in this study is a powerful interpretative framework that can identify the major reactions controlling the carbonate cycle. In the Chesapeake Bay, similar redox stratification can support varying intensity of carbonate dissolution, absent in 2017 or important as in 2018. In 2018, the "reaction driven" approximation also suggests an important role of the N cycle to consume alkalinity while, in 2017, nitrification was limited to 16% of the nitrogen mineralized. The summer anoxia observed in the Chesapeake Bay is characterized by an exceptionally high $\Delta TAex/\Delta DICex$

of 2.4 which has never been reported in anoxic water columns or sediment pore waters. The "reaction driven" approximation suggests it comes from sulphate reduction with almost complete hydrogen sulphide oxidation by MnOx followed by $MnCO_3$ precipitation. This interpretation is supported by the important manganese dynamics observed in this and previous papers (Oldham et al, 2017a; Sholkovitz et al 1992; Trouwborst et al, 2006). However, most of the reactions would have occurred in the sediment from which the components of the $\Delta TAex/\Delta DICex$ of 2.4 diffused into the bottom water with the redox front

during the set up of summer water column anoxia. Our study demonstrates that the Manganese cycle can have a strong impact on alkalinity, as it prevents $H_2S$ oxidation to $SO_4^{2-}$ and favour sulphur burial.

**Appendix 1: demonstration of equation (1)**

First, equation (A1) describes the behaviour of a solute, in case of turbulent diffusion mixing, with Ds the effective diffusion coefficient, superimposed on a chemical reaction of rate $v$ as described in equation (A2).

$$\frac{dC}{dt} = D_s \frac{d^2C}{dx^2} + \alpha_C \, v \qquad (A1)$$

$$\alpha_C C + \alpha_D D = \alpha_E E + \alpha_F F \qquad (A2)$$

Assuming steady state, *i.e.* dC/dt=0, and applying equation (1) to two different species sharing the same reactions at rate $v$ on each point and $\alpha_D \neq 0$, we can express the rate of the reaction as:

$$-\frac{v \, \alpha_C}{D_s} = \frac{d^2C}{dx^2} = \frac{\alpha_C}{\alpha_D} \frac{d^2D}{dx^2} \qquad (A3)$$

Equation (A3) is also true for any linear combination of a solute with a conservative element such as the salinity, S, because $\alpha_S = 0$, hence (with k any real):

$$\frac{d^2C + kS}{dx^2} = \frac{d^2C}{dx^2} + k\frac{d^2S}{dx^2} = \frac{d^2C}{dx^2}$$

By integration of equation (A3) with respect to x on a portion of space where $\alpha_C/\alpha_D$ is constant, it becomes equation 1.


**Appendix 2: validity of equation (2) when the two endmembers have similar concentrations**

From equation A3, and assuming that the rate $v$ can be described by a polynomial function of x, with a$_i$, any real coefficient, we obtain:

$$\frac{d^2C}{dx^2} = -\frac{\alpha_C}{D_S} \sum_i a_i x^i$$

That can be solve analytically and gives the following equation:

$$\frac{dC}{dx} = \frac{\alpha_C}{D_S} \sum_i \left[ \frac{a_i}{(i+1)(i+2)} \left(1 - (i+2)\, x^{i+1}\right) \right] + \Delta C$$

Here ΔC is the difference of concentration between the two endmembers. With similar reasoning on D and with ΔC = ΔD =0, we find equation (1) with G=0.

## Appendix 3: Additional figures

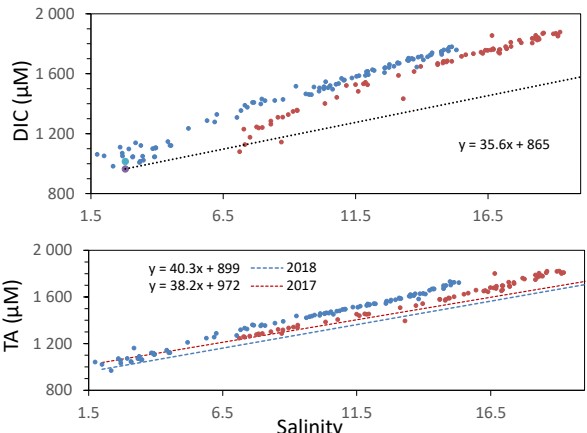

**Figure A1: Variation of Total Alkalinity (TA) during oceanic and river mixing. Dashed lines represent the theoretical DIC and TA if only mixing occurs.**

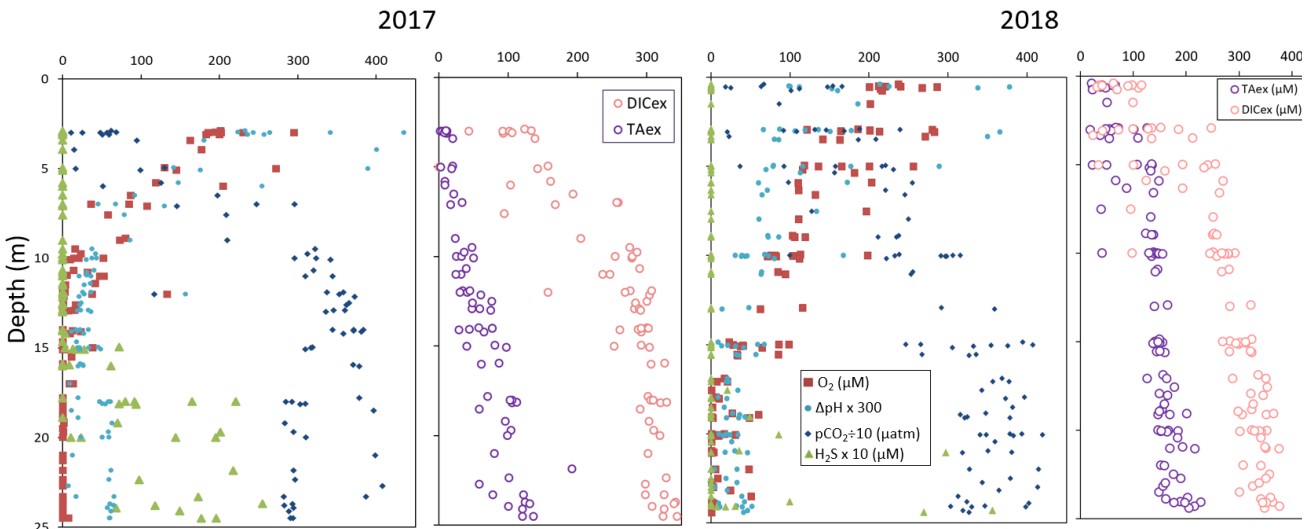

**Figure A2: Superimposed carbonate and redox chemistry profiles over 11 casts against depth.**


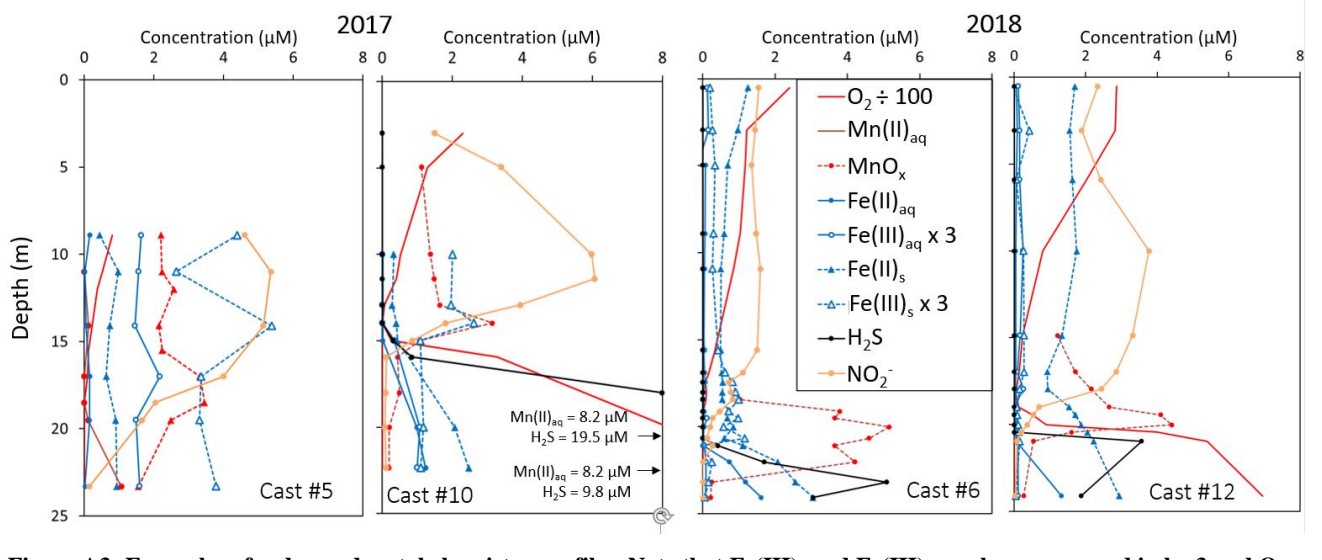

Figure A3. Examples of chemoautotroph density profiles. Note that Fe(III) and Fe(III) values are squared in the figure.

*Data availability*

The data used in this paper is available on request to the corresponding author.

*Author contribution*

ATC, ERE, JN, BMT and SJ performed the data analysis. ATC and SJ process the data. ATC, GWL, SJ and WJC interpreted the results. GWL, BMT and WJC get the funding. ATC wrote the paper with contributions from all authors.

*Competing interests*

The authors declare that they have no conflict of interest.

*Acknowledgements*

We gratefully acknowledge the support of the captain and crew of *R/V Hugh R. Sharp*. This work was funded by grants from the Chemical Oceanography program of the National Science Foundation (OCE-1558738 to GWL; OCE-1558692 to BMT and OCE-1756815 to WJC)

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
