# Peer review of "Influence of manganese cycling on alkalinity in the redox stratified water column of Chesapeake Bay"

_EGUsphere, 2022_

## Referee Comment (RC2)

General comments
Thibault de Chanvalon et al. have written a manuscript describing carbonate system dynamics in the Chesapeake Bay. The originality and novelty of the manuscript lies in the high-resolution measurements of iron and manganese species in Chesapeake Bay, of which the carbonate system dynamics have long been investigated by the senior author. However, the finding that Mn dynamics are most important for explaining the observed trend in ΔTAex to ΔDICex does not seem to be substantiated by the manuscript in its current form.

In particular, section 3.3 contains many assumptions that are not substantiated by measurements or modelling and in which a discussion on reaction kinetics is missing. For example, it is discussed in section 3.2 that slow reaction kinetics of nitrification may explain the lack of signal here, but there is no mention of kinetics impacting any of the (net) pathways in section 3.3. I think, however, that given the truly dynamic nature of the study site, kinetics may be key in explaining the observed trends, and that it may not be possible to do this using linear combinations of reaction equations and stoichiometry.

The introduction solely focuses on carbonate precipitation and dissolution, whereas the manuscript has a much broader focus and also investigate the role of the Fe and Mn cycles. I would encourage the others to expand the introduction by at least one paragraph explaining the role of other elemental cycles in alkalinity dynamics. This also makes a better transition towards explaining the aim of this study.

I do not advocate a combined results and discussion and especially in this manuscript it leads to confusion on what is interpretation and what is not. It shouldn't be too difficult to separate both sections. The conclusions on the important role of Mn at this site are rather speculative and contain references to other study that belong in the discussion.

The manuscript is generally written in a sloppy way. I stopped identifying typos already early on, simply because there are so many. So please do a proper check on this for a next version. Also, there are many exceptionally long sentences. This makes it difficult to following reasoning. At this stage, I'm afraid I cannot recommend anything else but a rejection. I do encourage the authors to rework the manuscript into a better one, because the data underlying the manuscript are interesting and of high quality.

Minor and technical comments
L. 12 (and other places): I do not like the term "alkalinity cycle" too much. This suggests the cycling of a particular element (e.g. C) whereas alkalinity dynamics are the result of the cycling of many different elements.
L. 19: What do ΔDICex and ΔTAex mean? Better not to use these abbreviations in the abstract.
L. 25: This is not a citation to the most recent global carbon budget. Also, estimates of the last decade mention that ca. 25% of anthropogenic CO2 has been absorbed by the oceans, not 33%.
L. 31: "shallow waters"
L. 32: "carbonate dynamics"
L. 39: not sure why specifically $HCO_3^-$ dynamics is used here. Isn't this more generically carbonate system dynamics?
L. 48: "sampling campaigns" (I stopped identifying typos here as there are too many)
L. 49: What is meant here, at 25-m water depth, or at a location with a water depth of 25-m? I assume the former, but please write more clearly (also in L. 54-57).
L. 65-85: this is not written in a very engaging way.
L. 91-92: Which CRMs?
L. 92-95: Why wasn't CO2sys used from the start? I'd say that it is common practice to use one of the packages for carbonate system calculations. And which equilibrium constants were used? Also, which other acid-base systems were taken into account? That matters for your conclusion that organic alkalinity is irrelevant in this study. In summary, this section severely lacks detail.
L. 103-110: The choice of the alkalinity freshwater endmember is extensively explained but I am not 100% convinced about it. Assuming there was some biological activity between S=0 and S=1.5, and

thus TA at S=1.5 is somewhat overestimated, how would that potentially affect the slope of the mixing line and consequently your excess TA?

L. 114: Section 2.3.2 does not deal with reaction stoichiometry; another title would be more appropriate

L. 116-140: Rather than defining a new equation and terminology, why not use either of the existing frameworks and corresponding terminologies of either Soetaert et al. (2007) (excess negative charge) or Wolf-Gladrow et al. (2007) (explicit conservative expression). The framework of Soetaert et al. (2007), specifically the definitions in section 3.3, seem as generic as the equations defined here because additional species can be included.

L. 141-146: It is completely unclear which reactions are referred to in this section. Specify / expand.

L. 149-152: I understand this choice but would be good to still show the plots versus depth as it is a more common way. Perhaps in supplementary information. Or show plots of salinity versus depth.

L. 152: "the processes" – which ones?

L. 152: "overall much lower salinity". Maximum values declined from ~20 to ~16, I wouldn't call that 'much lower'

L. 155: atmospheric pCO2 was likely higher than 400 uatm.

L. 162: I wouldn't call a zone with an oxygen concentration of less than 1 uM sub-oxic (I am in general not an advocate of this term), but rather anoxic

Fig. 1: I am not sure how correct it is to use a linear transformation for plotting for a non-linear variable like pH

L. 171-174: I can see this pattern in NO2 in the 2017 data but not really in the 2018 data. Do you have an explanation for this?

L. 175: This reads a bit odd. I think what you mean to say is that any O2 diffusing downwards would react with upwards diffusing Mn2+.

L. 184: "emerging picture"

Fig. 2: mistake in caption; this is nitrite, not nitrate

L. 200-207: If I remember correctly (I didn't look it up), in the model used by Cai et al. (2017) their results were explained by combining aerobic respiration with nitrification. Can the authors elaborate on the comparison with this study? In general, to me it seems surprising that no nitrification would take place.

L. 216: "as was observed for other years" – including 2017 or not?

L. 219 – 224: Good that saturation calculations were done here, although it would be good to actually present the (range of) values. Is there a logical source of calcite in this part of the Chesapeake Bay in 2017 that would support this hypothesis? (especially given what is written in L. 229 – 230) Also, are any analyses done on the type of algae that would contribute to PP?

L. 237: "never reported in the literature" – perhaps in estuaries or using this particular metric or ΔTAex/ΔDICex (although I am not even certain about this). But ratios of TA/DIC exceeding 2 have been discussed in earlier works.

L. 252 – 258: My main issue with this discussion is that there can be more fates of H2S than only discussed here, each having a different ΔTA/ΔDIC ratio. I am not sure that the choice of reactions discussed here and given in table 1 is properly substantiated, especially since no actual modelling has been conducted and since solid S species or MnCO3 have not been measured. As a result, the authors cannot state whether the formation of MnCO3 is actually important in the Chesapeake Bay. In fact, the authors state this to some extent themselves in L. 269-270.

L. 279: I don't understand the unit of ΔTAex here – or is something else meant?

L. 229 – 330: see comment before – I am not sure if this is really the case.

---

## Author Response (AR1)

Reviewer #1

This paper presents an original dataset of the carbonate system and major redox species in the water column of a stratified estuary with anoxic deep waters. The objective is to determine the main reactions that lead to a net production of alkalinity in the estuarine bottom waters. Based on an analysis of TA/DIC ratio and the stoichiometry of reactions, the authors conclude that MnO2 reduction followed by Mn carbonate precipitation are the two main reaction that can explain the observed trends in TA. I found many problems in this MS, including several small mistakes on the principles of the carbonate chemistry in the text, figures of poor quality not appropriate to describe the processes that are discussed in the text, a presentation aggregating results and discussion that makes the authors reasoning very hard to follow, and finally, a conclusion that appears speculative and not fully supported by the data. I had hard work reading the MS because of language problems and too many shortcuts all along, but I first thought it could be reconsidered after major revision because of the high quality of the data. However, when reaching the end of the discussion, I found the conclusions speculative and not fully based on appropriate quantitative statements, so I believe the analysis should be started over and the paper in its present form could be simply rejected, or at least revised in depth.

The "mistakes on the principles of carbonate chemistry" will be debated below.

The figures of poor quality not appropriate to describe the processes probably refers to the misunderstanding that all the data come form only one station

The figures of poor quality have been improved

The conclusions are based on observations clearly reported in the data i.e. ΔTAex/ΔDICex of 2.4 and the important MnO2 recycling and intensively discussed in the manuscript. It seems that the reviewer does not get the reasoning exposed in this ms, probably due to many shortcut and lack of detailed explanations provided. The revised version improves it a lot

Main problems:

1-presentation

Presentation of the results is confusing and the MS structure with a "result and discussion" section makes it worth.

Result and discussion have been written again and are now separated

Choice is made to present only plots of concentrations versus salinity aggregating two sampling periods. Readers cannot get a precise idea of the vertical structure of the water column in the estuary.

Each sampling period was presented in its own graph in Fig 1, there is no aggregation between the two sampling period. The revised caption state it clearly. Plot against salinity bring better view of the water column stratification since it follows the water masses based on their density. As proposed by the reviewer 2, an additional figure potting salinity and temperature against depth has been added (Figure 2); an additionnal Appendix (Fig. A2) plots $O_2$, pH, $H_2S$, $pCO_2$, TAex and DICex against depth has been added.

Conceptually, longitudinal gradients are mixed with vertical gradients without any consideration about respective mixing times.

Longitudinal gradient are not mixed with vertical gradient because all the samples are from the same station: Station 858. This information has been underlined in the revised version and is now repeated regularly including in section title, to avoid any misunderstanding.

A long and unnecessary discussion is made about the choice of the freshwater end-member values (which is not a real freshwater), although it appears that most of the salinity gradient studied here occurs vertically.

The word freshwater is not written in this manuscript, we clearly call it upstream estuary endmember. In the section 2.3.1 only 10 lines are now devoted to the endmember selection (versus 21 lines in the previous version) and more precision about the hypothesis required for the "reaction driven" interpretation were added. However, we think that the discussion about endmember selection is critical for the paper as the main data interpreted are DICex and TAex that vary based on the endmember selected. The definition of these endmembers can have impacts on all the dataset and requires careful definition.

We have no map of the estuary with sampling points.

It is because only one station has been sampled. The ms has been revised to highlight this important point.

No info about timescales, mixing of end members is analysed with little information about the timescale for the mixing to occur.

Mixing are estimated based on salinity that is conservative and assumption of steady state is now clearly explicit.

I was disappointed by the absence of real vertical profiles (some are in Fig A2, but not DIC and TA, pCO2, Sal, T, pH…) in the main text, although I understand the usefulness of the time composite salinity plot in Fig. 1 for the modelling and mass-balance purpose (assuming that the authors are able to demonstrate that the hypothesis behind such plot are valid)

An additional figure potting salinity and temperature against depth has been added (Figure 2); An additionnal Appendix (Fig. A2) plots $O_2$, pH, $H_2S$, $pCO_2$, TAex and DICex against depth has been added.

2- scientific content

see mistakes and imprecisions, in line by line comments

Stoichiometric model in fig.3 does not include the possibility for gas exchange to alter DIC and AOU, at least near the surface layer. Depending on pCO2 and O2 %sat values at the surface, gas exchange can alter the AOU/EDIC ratio away from 1. Deviation of the O2 / CO2 correlation can be due to carbonate buffering effect on CO2 but not on O2 (see e.g. in rivers: Stets et al. (2017), doi:10.1002/2016GB005578.).

Gas exchange cannot be taken into account in the "reaction driven" interpretation since gas exchange does not follow a fixed stoichiometry but depend of the extent of a disequilibrium. We now explicitly point out this assumption and focus the discussion below 3-meter depth.

In fact, because the salinity and redox gradients are vertical at the study site, there is no need to always consider the gas exchange and its impact on DIC. Using the pCO2 value at the surface, it would be possible to calculate the change in DIC due to invasion of atmospheric CO2 in the first meters of the water column during the representative mixing time, and show that this change is negligible.

Water atmosphere exchange have been extensively investigated by Chen et al. 2020 on the Chesapeake Bay. In our case, it would be only possible for the 2018 campaign but is beyond the scope of our study. At 3-meter depth and below it can be considered negligeable.

The main problem I see concerns the conclusion that Mn oxide reduction coupled to Mn carbonate precipitation is the "key" mechanism.

First it is not clear in the MS what are the respective roles of Mn and Fe, both appear in the title and abstract, but only Mn is considered in the stoichiometric approach.

Both are considered in the stoichiometric approach (see Table 1). However, since our dataset indicates that only the manganese seems important in our study case, we modify the title.

Second, the argumentation on the predominance of Mn reactions on the production of alkalinity is based only on the TA/DIC slope of 2.4 observed between the suboxic zone and the sulfidic zone (Fig. 3). However, there are many processes and reactions potentially occurring in this transition zone and the slope of the TA/DIC ratio does not only depend on the nature of the reactions as the authors analyse, but also on the intensity of the reactions and on the mixing intensity by turbulence and vertical transport between the two layers.

When the reactions occurring in the suboxic or sulfidic zone during a water mass journey are combined together, it results overall in a few possible budget reactions that we summarized in the Table 1. Such approach has been already used in porewater by Rassmann et al. (2020) or in the water column by Hiscock and Millero (2006). We agree that even in these two publications, the assumption require for such approach where not explicitly described. Therefore we added a section 3.2.1 to trigger the condition where this approach (called "reaction driven" interpretation) is valid. The beginning of the discussion (section 4.1) focus on the applicability of the "reaction driven" interpretation on the station 858 in particular:

"Assuming 1) that mixing is efficiently described by turbulent diffusion mixing, 2) that the measured concentrations correspond to a steady state, 3) that the concentration at the starting point does not vary with time and 4) that the samples are isolated from atmospheric exchanges; the "reaction driven" interpretation (section 2.3.1) permits interpretation of the concentration changes as a linear combination of the stoichiometry of several chemical reactions (equation 5)."

Indeed, at the top of this suboxic-sulfidic gradient some TA consuming processes by secondary reaction not involving organic matter can occur. So the presented analysis considering the TA/DIC slope variations with the stoichiometry of primary reactions followed

by secondary reaction such as CaMn precipitation does not account for secondary reactions that may decrease the TA without changing the DIC.

The "reaction driven" interpretation does not describe what is currently occurring at a given location but it describes the dominant reactions that modify a water mass chemistry during its overall journey since its equilibrium with the initial endmember. So it does account for secondary reactions that decrease the TA without changing the DIC. In particular, if $H_2S$ diffused and is reoxidized by oxygen forming $SO_4^{2-}$ the TA will decrease without changing the DIC. Therefore, the signature of a water mass submitted to sulfate reduction then to oxygenated oxidation will correspond to the signature of sulfate reduction plus the signature of oxygenated oxidation. It results in a signature similar to aerobic respiration (change of TA and DIC) that is taken into account in our description. We hope that the new version of the manuscript explain it better now.

Finally, the MS itself reveals the weakness of the conclusion about the importance of Mn reactions: L310 "Based on an average concentration of 20 μmol g-1 of Mn in suspended particles, the 88 μM of MnO2 would require a suspended material concentration of about 4.4 g L-1, which is again one or two orders of magnitude higher than the 0.01 – 0.1 g L-1 usually found in the Chesapeake Bay."

We took special care to explicitly describe the boundary of our model and push it into its limitation, including saturation index calculation and mass balance calculation to validate the "reaction driven" interpretation. We explain the difference between Mn pool and TA pool by the role of sediment reactivity.

Line by line comments

Abstract

« burial of carbonate which modulates the ability of the ocean to trap anthropogenic CO2. ». Not clear if you refer to build up of alkalinity in the ocean, which is indeed a sink of atmospheric CO2 or burial of CaCO3 which leads to degassing of CO2 (Buffer factor 0.7, See works by Frankignoulle and Gattuso end of 90s in coral reefs). burial of carbonate would not trap anthropogenic CO2.

The citation should include the subject of the verb : "The coastal alkalinity cycle controls the global burial of carbonate which modulates the ability of the ocean to trap anthropogenic $CO_2$", to clarify the sentence we change it into: "The alkalinity dynamic in coastal environments controls the global burial of carbonate and modulates the ability of the ocean to trap anthropogenic $CO_2$"

L15: according to the profiles, how was NEP distributed? Positive at the surface and negative below? Yes

You mention carbonate dissolution, but no precipitation occurs for instance in blooms at the surface? Precipitation was not visible based on the "reaction driven" approach and was probably only of minor importance, if any. The observed zone of primary production is at the subsurface as now clearly state

L28 "weathering" rather than "erosion" changed

"enrichment not associated with a Ca2+ enrichment, in contrast to the HCO3- released from continental erosion (preponderant at thousands to a million year scale, Urey, 1952)." HCO3- also comes from dissolution of continental rocks others than carbonate rocks (100% atmospheric CO2), and thus without Ca2+ enrichment.

It seems that the reviewer refer to MORB degassing, whose part can occur during igneous rock weathering, we change the sentence in . "At the century time scale, atmosphere-ocean exchanges result in oceanic $HCO_3^-$ enrichment not associated with a cationic enrichment, in contrast to silicate or carbonate weathering (preponderant at thousands to a million year scale, Urey, 1952)"

L40 "with a high vertical resolution (down to 10 cm)." a pity we cannot see these high-resolution profiles in the MS, at least some XXX Some examples are presented in the Appendix Figure A3

Introduction is focussed on CaCO3, but the paper is mostly based on an analysis of TA/DIC ratios. Better introducing the principles of TA/DIC ratio analysis would be helpful. It is done in the section 2.3.1 of the new version.

L49: eleven or "a dozen"? "a dozen" has been removed from the manuscript due to its lack of precision

L62 define DI done

L93 "an excel sheet implemented with values from (Millero, 1995)" > which carbonate and bicarbonate dissociation constants and solubility coefficient of CO2 ? We clarify this part : "The pCO2, calcite saturation and TA were calculated from measured DIC and pH via CO2sys program using Cai and Wang (1998) constants."

L100 "TA and DIC are conservative during mixing" for DIC, this is true only if no gas exchange occurs, which is the case vertically in a water column, below a certain water depth that could be calculated with the data and a simple gas exchange parameterization. We now clearly precise that the discussion is focussed below 3 meter depth away from atmospheric exchanges

L108 "Such changes were not necessary for DICex calculation." Why? The choice of the freshwater TA & DIC end-members looks arbitrary. Please better explain. In theory, the sensitivity of calculated DICex and TAex to the values in the freshwater end-members can be calculated

The DIC versus salinity plot has been added in the Fig. A1. We wrote again the explanation: "the oceanic endmember was the one proposed by Su et al. (2020a) for August 2016 campaigns. Oceanic endmember varies mainly with season (Cai et al., 2020) and a change of 50 μM results in 5% uncertainty on the slope of the mixing line. Large variations exist in the

upstream estuary endmember mainly due to changes of weathering intensity and riverine discharge (Meybeck, 2003; Joesoef et al., 2017) and a one-off endmember has to be determined by fit with the in situ measurements at the lowest measured salinity (Fig. A1)."

We also add a calculation for the uncertainty:

"The uncertainty of ΔTAex/ΔDICex is equal to the sum of the relative uncertainty of ΔTAex and ΔDICex. Posing ΔTA the change of TA measured, ΔS the change of salinity and sml_TA, the slope of the mixing line for TA, we have ΔTAex = ΔTA – sml_TA * ΔS. Uncertainty on ΔTA and ΔS are negligeable face to the relative uncertainty of slope_ml and posing δ(x) the incertitude on x, we get:

$$\frac{\delta\,(\Delta TAex/\Delta DICex)}{\Delta TAex/\Delta DICex} = \frac{\delta(\Delta TAex)}{\Delta TAex} + \frac{\delta(\Delta DICex)}{\Delta DICex} = \frac{\delta(sml\_TA)}{sml\_TA} + \frac{\delta(sml\_DIC)}{sml\_DIC} = 0.1 \qquad (6)$$

"

L111 "the uncertainty of our description" Awkward formulation changed

L151 "While direct plots against depth generate noisy profiles that are less informative, plots against salinity provide consistent information about the processes.". In a section called "water column stratification", one would expect to see at minima T and S profiles versus depth.

We add a figure with the temperature and salinity against depth.

Why "noisy profiles"? are they altered by the sampling procedure? How can profiles be noisy versus depth but not versus salinity? Are salinity profiles "noisy"? Does the sampling keep the stratification intact? Is the noise due to heterogeneity induced by tidal currents despite samples being taken at tidal slack?

It is now explained: "Plots against depth generate noisier profiles are shown in Appendix 2 while plots against salinity follow the water masses." Indeed, between each sampling the water masses move due to tidal and river currents and are not located exactly at the same depth from one cast to another.

L152 > "River" flow changed

L158 "Below, with increasing depth, an important increase of pCO2 accompanying the decrease of O2, pH and temperature is visible." In fact, readers cannot see anything "visible" "with increasing depth", only Fig A2 reports vertical profiles, and no S, T, pH, O2, pCO2 are shown.

The new result section clarifies these points.

In addition, it looks that some 2017 profiles start only at 8m, why? In Fig. 1 the surface "PP" layer is referenced as 2 meters depth, does this mean that the surface layer sampling includes only some of the stations? Why such strategy? Why no systematic surface sampling? This needs clarification

The main goal was not to describe the air-water exchange. Thus we show the data when available (below 3 meter depth in 2017 and below 0.8 meter depth in 2018) but we focus the discussion on the redox gradient.

L159: "A relatively invariable low O2 zone (called ILO in Fig. 1) is here defined by the depth invariance of O2 concentrations, and 160 corresponds to a concentration of about 30 μM in 2017 and 110 μM in 2018. Other species are also relatively stable for this depth such as pCO2, at about 2500 μatm in 2017 and 1800 μatm in 2018, and pH, about 7.3 in 2017 and 7.4 in 2018." Readers have no idea what depth you are referring to. There is no figure versus depth for these parameters

We change "depth invariance" by "salinity invariance".

"The main changes between the two campaigns correspond to a greater oxygen penetration in 2018, preventing nitrite accumulation and to the appearance of a surface layer (with salinity below 3) that stands above the primary production zone in 2018." Readers are lost not only because figures are not showing what you are referring to, also because you show and discuss the data at the same time.

Result and discussion are now separated

L172 "Because of the presence of oxygen, the NO2- production would be more likely associated with nitrification of the NH4+ diffusing upward rather than denitrification despite the possibility of reducing conditions occurrence in micro niches." No NH4 and NO3 data are shown, it looks speculative.

Replaced by "Below, a low oxygen layer with invariant concentration of most species survey (the ILO zone) is characterized by significant nitrite accumulation in 2017 probably due to oxidation of $NH_4^+$ diffusing upward (Fig. 3)."

L178 "The MnOx decrease fits perfectly to the Mn2+ increases in sulfidic conditions (Fig. 2)". sorry, I could not find this perfect fit in the sulfidic zone in Fig.2

Replaced by "the MnOx disappearance corresponds to the $Mn^{2+}$ increases"

L180 "efflux" > "flux" the sentence has been removed

End of page 7: you are discussing analytical aspects in a section about vertical stratification. This section is very difficult to follow. Rewrite. done

L192 "However, due to river mixing with ocean waters…" the paper appears very confusing when it aggregates all mixing processes, spatial and time scales: longitudinal and vertical, seasonal. What are the typical vertical and longitudinal mixing times? I guess if the vertical structure is stable over time as the authors write, then vertical mixing is slow and the described geochemical reactions occur at timescales of months to year? This is what justifies

the use of a single plot that aggregates the two seasons? Then the longitudinal mixing with seawater is not sampled here (salinity > 20) but it could be shorter? Mixing of buffered marine water with anoxic bottom waters (and thus reoxidation reactions) at low river flow and mixing with surface fresher waters at high river flow?

As precised previously, there is only vertical variation reported in our publication for two summer in August 2017 and August 2018. Season are not aggregates. The new version clarifies this point.

L197 "This offset is within the uncertainty of the endmember calculation even if slight DICex background enrichment has been modelled (Shen et al., 2019) resulting from faster atmospheric equilibration of O2 than CO2 after respiration reactions." Not clear what you mean here: what has DICex enrichment to do with different rates of CO2/O2 atmospheric exchange? Please explain. This is a classical problem when results are mixed with discussion. The amount of data presented and the relative complexity of the geochemical analysis make the combination of result and discussion sections very difficult to follow. This side interpretation has been removed to discard any discussion about air-water exchange that were not well constrained in our sampling design.

What is the point discussing these values of freshwater end-member? all the reactions described in the paper occur at salinity >1.7 or 7.1, so this salinity value can be used as end-member. Extrapolating all TA values until Sal 1.7 (high river flow) as done in the MS is ok if vertical mixing time , no need for a long text about the choice of FW end-member, same for SW. Describe in Mat & Met done

Fig 3 panel c: arrow direction "CO2 uptake" would decrease DICex, not increase

The arrow has been removed to focus on redox reaction in the water column, but from a water column point of view, uptake suggests that this is an uptake from atmosphere into the water column, so it increases DICex.

L200 "Interestingly, the relative changes of DICex and TAex, further named $\Delta$TAex and $\Delta$DICex, does not depend on the endmember calculation and their ratio presents much lower uncertainties (about 0.1) facilitating their interpretation". This could be partly transferred to the Mat & methods section. A scheme in a suppl. figure could also help to define the variables along the vertical salinity gradient. We add a figure in the section 2.3.1 that explicitly describe the condition for the "reaction driven" interpretation

L200 "In 2017, TAex stayed almost constant up to the oxic zone (Fig. 3a)" we cannot see the "oxic zone" in Fig 3a, only guess it. Result section could show depth profiles and discussion the salinity plot

We can easily infer where is the oxic zone looking at a Figure plotting apparent oxygen uptake.

"$\Delta$TAex/$\Delta$DICex ratio of 0.1 ± 0.1 which indicates a net aerobic respiration (AR)" not only respiration, also primary production assimilating NH4+. What is a "net" aerobic respiration?

We agree that both AR and PP are characterised by the slope $\Delta$TAex/$\Delta$DICex =0.1, but since in our case it result in a increase of TAex it result in net AR (AR>PP). It is now precised in

section 4.1: "This interpretative framework describes the vertical stratification of the water column as the journey of a water mass slowly mixed deeper and deeper and whose DIC and TA are progressively enriched by all chemical reactions they undergo. Accordingly, this interpretation does not identify reactions with minor impact on the carbonate cycle or reactions cancelled later during the journey, for example, PP is frequently cancelled by similar amount or excess of AR."

What has the discussion in L200-L220 to do with "river flow control", the title of the section? The MS needs to be reorganized. done

L210 "Finally, for 2017, despite $pCO_2$ being below atmospheric saturation at about 2 m depth (Fig. 1), the possible $CO_2$ invasion does not significantly modify the observed $\Delta DICex/\Delta AOU$ signal at the shallowest depth sampled." I agree that gas exchange is a slow process compared to PP and AR. However, gas exchange still occurs and it affects DIC/AOU ratio with a slope still close to 1

Because gas exchanges are proportional to the disequilibrium it does not result in a fix DIC/AOU ratio and is therefore not considered in the "reaction driven" interpretation.

L215 "In 2018, this surface water history did not repeat as fresh and light water masses brought by the exceptional flood drastically modified the carbonate system equilibrium. First, a low salinity layer with $pCO_2$ at 1000 µatm overlays the primary production layer (Fig. 1)" We really need to see the most relevant vertical profiles…. Or isolines. Now available in Figure A2

L217 "Just below the air-sea interface, the lock down of atmospheric exchanges by the law salinity layer produces supersaturation of trapped $O_2$ (Fig. 1, for S between 3 and 4)." Exhausting to follow. Contrarily to the authors, the reader has not seen the vertical profiles before. Law salinity > low salinity

"In Fig. 3a and 3b, this process translates into a vertical distribution at $DICex = 40$ µM associated with negative AOU and slightly positive TAex." I see no "vertical distribution" in Fig 3. Negative AOU is nothing special in surface productive waters

An arrow show the vertical distribution of TAex and AOU. It is not particularly special, it just reflects the reaction CD-AR (table 2)

L219 "This original signature can be modelled by the combination of simultaneous carbonate dissolution (CD) and PP fuelled by $NH_4^+$, in equal proportion and would result in no DICex, only TAex production (see Table 2); the carbonate dissolution buffers the DIC consumption produced by PP. The $Ca^{2+}$ concentrations observed by Su et al. (2021) and during the 2018 cruise (data not shown) vary linearly with salinity i. e. $[Ca^{2+}] = 0.282 S + 0.4$ in mmol L-1. Assuming similar behaviour in 2017, calculations show that the whole water column is undersaturated with respect to calcite and validates the possibility for CD." This seems speculative (no calcite saturation value is shown); in general, PP increases the pH and favours $CaCO_3$ precipitation rather than dissolution. One could also say that if $Ca^{2+}$ is conservative, then little or no precipitation/dissolution occurs.

Thanks to this comment, we take a more careful analysis to the $Ca^{2+}$ profile. It came out that "The $Ca^{2+}$ concentrations observed by Su et al. (2021) and during the 2018 cruise (data not

shown) vary linearly with salinity. Assuming similar behaviour in 2017, calculations show that the whole water column (except 4 samples from the PP zone) is under saturated ($0.36 < \Omega_{cal} < 1$; mean=0.68) with respect to calcite in 2018, while undersaturation is only valid below S=10 in 2017."

In this part, we have to explain an increase of TA, a decrease of AOU and no DIC changes. Which fit with CD and PP. In this layer, the calcite saturation increase up to saturation for the 4 points with higher pH. It is thus possible to have a bit of precipitation in a thin layer, but it does not overwhelm the signature of carbonate dissolution that occur previously in these water mass.

$Ca^{2+}$ concentration vary linearly with salinity in our sample, but, compare to the mixing line between estuarine and oceanic water it corresponds to a $Ca^{2+}$ excess of up to 200uM.

L220-230 are difficult to follow; this section starts identifying some preponderant reaction at the top of the water column (oxic condition, what about reoxidation of reduced species diffusing from below?), but the following section is entitled "Identification of preponderant reactions". The paper needs a better organisation. We hope that the new organisation make the reading easier.

There seems to be a mistake in reaction SR-O In table 1, H2SO4 appearing on both sides of the arrow. If you eliminate H2SO4, then the reaction is aerobic oxidation. In fact you cannot combine two reactions when one occurs only in oxic condition and another only in anoxic condition.

It is sure we can combine two reactions occurring in different conditions, as soon as a mixing makes possible the transfer of the solutes at similar rate which is the case with turbulent diffusion.

L235 remove "3.3. Sediment control" done

L238 "cannot be explained by most typical chemical reactions such carbonate dissolution (CD), aerobic respiration (AR), CO2 uptake or primary production (PP = -AR)." I though this section concerned anoxic conditions. This is confusing

Removed

L241 "Moreover, SR alone underestimates the importance of the H2S oxidation pathway." Not sure what you mean here, please reword

The new version is :

Moreover, SR alone underestimates the importance of the $H_2S$ oxidation pathway that can consume all the alkalinity produced during SR. For example, SR follow by oxygenated oxidation results in $\Delta TAex/\Delta DICex/\Delta AOU$ signature equal to AR only. In the Chesapeake Bay, $H_2S$ oxidation is critical since no $H_2S$ is measurable in the suboxic zone while the gradient at the sediment/water interface indicates high $H_2S$ sedimentary efflux (Fig. 3).

L246 "Middelburg et thal., 2020" many typos in the MS changed

L248 "corresponds to the uncharged species produced, mostly in solid or gaseous phases" be precise… you mean N2 by denitrification and FeS ?

it can be $N_2$, $N_2O$, $CO_2$, FeS $FeS_2$ FeOOH $MnO_2$ $CaCO_3$ …. we stay general to not be too specific and precise later in the ms and in Table 1

L257 "a non-charged species." > specie this sentence has been removed

L260 reasoning on the importance of N based only on NO2- data looks speculative if no NO3- / NH4+ are shown

It is now stated: Therefore, in the absence of nitrate, oxygen and $H_2S$,

Fig4: show all units this figure has been removed

L263-266 Suddenly, the authors mention a "monthly timescale" without any apparent reason for that. Replaced by at steady state

L267 " "important" stock concentrations at a monthly timescale (with concentrations that frequently exceed 1 mM in anoxic porewater)." Why porewater? The study does not deal with sediment changed into anoxic water

L272 "represent the main expected respiration processes" please better explain why the 2 mentioned reactions are expected to predominate. Why Mn more than Fe? Because the admitted form of Fe is FeS or $FeS_2$ that can be only produced with sulphide.

L270-285 contains many shortcuts and language imprecisions and hardly convince the readers that the mentioned reactions are preponderant. The analysis appears almost only qualitative, based on TA/DIC ratios, not quantitative based on concentrations and mass-balance

"The reduction of HNO3 down to NH3 is not detailed but would result in almost similar alkalinity changes: 1.15 for NH3 production (DNRA) versus 0.95 for N2 production." Instead of "alkalinity changes", do you mean TA/DIC ratio? This sentence has been removed

"The only solid form of Mn(II) is MnCO3, since MnS is negligible" why should Mn(II) be solid? Unclear. In fig 2 max Mn(II) concentration is about 8microM in the bottom layer, how does this contribute stoichiometrically to the 100 microM increase of TA? The discussion based on mass balance is now presented in the dedicated section 4.5

L275 "FeCO3 production would produce a very similar reaction as MnCO3 production; the latter, more common, is favoured in this simple description." What "simple description" are you referring to here?

The above remarks correspond to a section that has been re written and is now:

To build a pool of candidate reactions for the fitting, first, dissolved species at too low concentration (e.g. Mn2+aq, Fe2+aq) to be a net reagent to affect the carbon cycle at steady state are not taken into account. These species are usually recycled rapidly and hold a role of catalyser or electron shuttle between other redox species. Second, many mineral expected at low concentration or thermodynamically not favoured and their associated reactions are

neglected (e.g., iron phosphate, ferrous or manganous oxide, sulphur clusters, MnS, FeCO3, adsorption processes, reverse weathering). Therefore, only aqueous species with important stock concentrations (that can exceed 1 mM in anoxic water) are taken into account, i. e., $SO_4^{2-}$, $Ca^{2+}$, $H_2S$, $NH_4^+$ together with gaseous ($N_2$, $CO_2$) and main solid phases ($FeS_2$, FeS, S0, $MnCO_3$, FeOOH, $MnO_2$). Third, many combination of carbon remineralisation reaction with a re-oxidation reaction are equal to another remineralisation reaction. As an example, SR follow by H2S oxidation with oxygen is equal to aerobic respiration.

L277 "minimum required amount of sulfate reduction" Awkward formulation improve language this sentence has been removed

L280 "After sulfate reduction, H2S can also accumulate in the water (SR reaction) or be oxidized back to SO42- (SR-O is detailed as an example)." Yes indeed. However, H2S oxidation by O2 will lead to zero delta DIC and negative delta TA, and would result in an increase in the observed dTA/dDIC slope without the necessity of involving Mn secondary reactions. If H2S is totally reoxidized by O2, then the overall delta TA is null.

If H2S is totally reoxidized by O2 after sulfate reduction, the overall $\Delta$TA is 0.15 due to ammonium release, $\Delta$DIC is 1 and $\Delta$AOU is 1 exactly as if it was aerobic respiration (Table 1). It is not involving Mn secondary reaction but is not able to explain the $\Delta$TAex/$\Delta$DICex of 2.4.

L285. Readers need to know to what redox zone you are refering to. This 2.4 slope concerns only the suboxic-sulfidic transition zone. In general, the paper needs a better organization and more detailed discussion.

L310-320 seems speculative and is not convincing: in the water column the quantities of Mn is not sufficient to validate the stoichiometric model. Why should the reaction occur in the sediment, if the 2.4 TA/DIC slope concerns the bottom of the water column and not the sediment porewater? The reaction should occur in the sediment because the quantities of Mn in the water column is not sufficient to validate the stoichiometric model. Because the Mn product is solid it would have stay in the sediment while the $\Delta$TAex/$\Delta$DICex fingerprint would have been able to diffuse out of the sediment into the water column with other reduced species as the redox front move up when the summer start.

"The sedimentary solid Mn stock is about 10 mM" per square meter, per kilogram?

mM is mmol $L^{-1}$ per liter of porewater

"which largely exceeds the 88 μM required to produce the 100 μM TAex increase." The TA increase occurs in the bottom waters, if you want to relate it quantitatively to the sediment content, you should not only compare concentrations, but rather upward and downward fluxes in the bottom layers and at the sediment-water interface.

We agree that a direct flux measurement will give more convincing argument. Therefore we can only say that it is possible. We try to find the more likely explanation to an observable. Accordingly we use conditional formulation .

"Therefore, the Chesapeake Bay sediment is particularly rich in manganese and could host important SR-MnC reactions in the superficial pore water whose soluble products diffuse up to the water column during summer could bear with them the high $\Delta TAex/\Delta DICex$ signature observed."

L320-326 are disconnected from the rest of the paper

This section proposes to upscale the process observed in the Chesapeake Bay at the global scale.

Reviewer #2 :

General comments

Thibault de Chanvalon et al. have written a manuscript describing carbonate system dynamics in the Chesapeake Bay. The originality and novelty of the manuscript lies in the high-resolution measurements of iron and manganese species in Chesapeake Bay, of which the carbonate system dynamics have long been investigated by the senior author. However, the finding that Mn dynamics are most important for explaining the observed trend in ΔTAex to ΔDICex does not seem to be substantiated by the manuscript in its current form.

In particular, section 3.3 contains many assumptions that are not substantiated by measurements or modelling ….

We still believe that an approach based on linear combination of reaction stoichiometry is valid. A new section (2.3.1 *Identification of biogeochemical process from scatter plot: the hammer, the bow and the spear*) focusing on possible interpretations of a scatter plot of two species now describes more precisely the assumption required.

…and in which a discussion on reaction kinetics is missing. For example, it is discussed in section 3.2 that slow reaction kinetics of nitrification may explain the lack of signal here, but there is no mention of kinetics impacting any of the (net) pathways in section 3.3. I think, however, that given the truly dynamic nature of the study site, kinetics may be key in explaining the observed trends, and that it may not be possible to do this using linear combinations of reaction equations and stoichiometry.

We agree that kinetic consideration may be a key in explaining why such a reaction is preponderant over another one, thermodynamic being the second key. It is implicitly taken into account in our model by the coefficient of the linear combination, the section 2.3.1 now clearly explains how. By the way, this stoichiometric approach has already been applied to water column without extensive justification (*e.g.* Hiscock and Millero, 2006).

The introduction solely focuses on carbonate precipitation and dissolution, whereas the manuscript has a much broader focus and also investigate the role of the Fe and Mn cycles. I would encourage the others to expand the introduction by at least one paragraph explaining the role of other elemental cycles in alkalinity dynamics. This also makes a better transition towards explaining the aim of this study.

We added new section in the introduction

I do not advocate a combined results and discussion and especially in this manuscript it leads to confusion on what is interpretation and what is not. It shouldn't be too difficult to separate both sections.

Result and discussion have been separated and write again

The conclusions on the important role of Mn at this site are rather speculative and contain references to other study that belong in the discussion.

Conclusion has been deeply reworked in order to focus on demonstrated results and clearly specify what belong to hypothesis, in particular to the "reaction driven" interpretation.

The manuscript is generally written in a sloppy way. I stopped identifying typos already early on, simply because there are so many. So please do a proper check on this for a next version.

Also, there are many exceptionally long sentences. This makes it difficult to following reasoning. At this stage, I'm afraid I cannot recommend anything else but a rejection. I do encourage the authors to rework the manuscript into a better one, because the data underlying the manuscript are interesting and of high quality.

Minor and technical comments

L. 12 (and other places): I do not like the term "alkalinity cycle" too much. This suggests the cycling of a particular element (e.g. C) whereas alkalinity dynamics are the result of the cycling of many different elements.

This term has been changed into "The alkalinity dynamic in coastal environments … »

L. 19: What do ΔDICex and ΔTAex mean? Better not to use these abbreviations in the abstract.

These abbrevations have been deleted and replace by : "In oxygen depleted waters, 2.4 mole of DIC is produced per 1 mole of TA production. This substantial DIC increase relative to TA has not been previously reported in the literature, and is consistent over the two years"

L. 25: This is not a citation to the most recent global carbon budget. Also, estimates of the last decade mention that ca. 25% of anthropogenic CO2 has been absorbed by the oceans, not 33%.

More recent reference was add (Friedlingstein et al., 2022) instead of (Friedlingstein et al., 2019), in which they precise that 29.5+-5% of anthropogenic CO2 has been adsorbed by the oceans. It is now precised "30%"

L. 31: "shallow waters" corrected

L. 32: "carbonate dynamics" corrected

L. 39: not sure why specifically HCO $_{3-}$ dynamics is used here. Isn't this more generically carbonate system dynamics? We delete this part of the sentence

L. 48: "sampling campaigns" (I stopped identifying typos here as there are too many) corrected. All the typos have been review again to limit the errors.

L. 49: What is meant here, at 25-m water depth, or at a location with a water depth of 25-m? I assume the former, but please write more clearly (also in L. 54-57). It is now written: "eleven profile casts were conducted in a unique station in the Chesapeake Bay with a water depth of 25 m »

L. 65-85: this is not written in a very engaging way.

L. 91-92: Which CRMs? It is now precised

L. 92-95: Why wasn't CO2sys used from the start? I'd say that it is common practice to use one of the packages for carbonate system calculations. And which equilibrium constants were used? Also, which other acid-base systems were taken into account? That matters for your conclusion that organic alkalinity is irrelevant in this study. In summary, this section severely lacks detail.

This section was written again :

"The $p$CO$_2$, calcite saturation and TA were calculated from measured DIC and pH via CO2sys program using Cai and Wang (1998) constants. The measured TA was found highly correlated to the calculated TA ($r^2$ = 0.995 and 0.998, slope = 0.995 and 1.017 for 2017 and 2018 campaign respectively) and their difference was always below 30 µM with an average of 7.5 µM for 2017 and of 22.2 µM in 2018. These results suggest low contribution of non-carbonate species (e.g. nitrite, ammonium or organic matter (Cotovicz Jr. et al., 2016)) and measured TA was used for the interpretation."

L. 103-110: The choice of the alkalinity freshwater endmember is extensively explained but I am not 100% convinced about it. Assuming there was some biological activity between S=0 and S=1.5, and thus TA at S=1.5 is somewhat overestimated, how would that potentially affect the slope of the mixing line and consequently your excess TA?

There can be intense biological activity or weathering between S=0 and S=1.5, or even upstream in the river; what matter is the speed of variation of the endmembers. Especially if the riverine endmember changes faster than the water flow through the estuary. In our case it is safer to take an endmember at S=1.5 that represent a large pool of water in the Chesapeake Bay (with significant inertia) rather than the endmember at S=0 in the Susquehanna River that flow continuously and could change rapidly. The section 2.3.1 now clearly stipulates

"The upstream endmember is not a river endmember (Su et al., 2020a) but corresponds to a salinity above 1.5 preventing any interpretation for biological activity in the fresh water part of the estuary (Meybeck et al., 1988). However, it corresponds to a larger water mass pool, less sensitive to short term changes and thus more likely to satisfy the condition of stability of the endmember."

The uncertainty related to the change of endmember in now presented in section 2.3.1:

The uncertainty of ΔTAex/ΔDICex is equal to the sum of the relative uncertainty of ΔTAex and ΔDICex. Posing ΔTA the change of TA measured, ΔS the change of salinity and sml_TA, the slope of the mixing line for TA, we have ΔTAex = ΔTA – sml_TA x ΔS. Uncertainty on ΔTA and ΔS are negligible to the relative uncertainty of slope_ml and posing δ(x) the uncertainty on x, we get:

$$\frac{\delta\,(\Delta\mathrm{TAex}/\Delta DIC\mathrm{ex})}{\Delta\mathrm{TAex}/\Delta DIC\mathrm{ex}} = \frac{\delta(\Delta\mathrm{TAex})}{\Delta\mathrm{TAex}} + \frac{\delta(\Delta DIC\mathrm{ex})}{\Delta DIC\mathrm{ex}}$$
$$= \frac{\delta(\mathrm{sml\_TA})}{\mathrm{sml\_TA}} + \frac{\delta(\mathrm{sml\_DIC})}{\mathrm{sml\_DIC}} = 0.1 \qquad (6)$$

L. 114: Section 2.3.2 does not deal with reaction stoichiometry; another title would be more appropriate

We now precise :

"From Eq. (12), one can easily deduce the changes of alkalinity from any reaction stoichiometry as soon as the bearing charges at pH = 4.5 are known."

L. 116-140: Rather than defining a new equation and terminology, why not use either of the existing frameworks and corresponding terminologies of either Soetaert et al. (2007) (excess negative charge) or Wolf-Gladrow et al. (2007) (explicit conservative expression). The

framework of Soetaert et al. (2007), specifically the definitions in section 3.3, seem as generic as the equations defined here because additional species can be included.

Reviewer #2 considers that the terminology proposed by Soetaert et al (2007) is as generic as the former equation (4), now improved into the equation (12). The Soetaert et al (2007) publication was a fantastic source of inspiration, nevertheless their definition of alkalinity is based on the excess of negative charge plus/minus six specific chemical species:

$$TA = \Sigma[-] + \Sigma\ NH_3 - \Sigma\ NO_3 - \Sigma\ NO_2 - \Sigma\ PO_4 - 2\ \Sigma\ SO_4 - \Sigma\ F \qquad\qquad (A1)$$

With $\Sigma$ [-] being the sum of total charge of all the acid-base species. Which requires first to identify all acid-base species and second to know the exact charges they bear at the sample pH (Fig 1B from the publication can help you) which is much less synthetic than our equation (4) from the original manuscript and than the equation 12 of the current manuscript. $TA = \Sigma\ z^{pH=4.5}$

For example, the equation A1 does not allow the reader to easily understand how "new" species (such as reduced FeS clusters or DOM) will change the alkalinity, which is what our equation (12) does.

L. 141-146: It is completely unclear which reactions are referred to in this section. Specify / expand.

All this section has been expanded in section 2.3.1 to clearly justify the use a linear combination. Moreover, it is now precised "A limited number of reactions is selected as candidates based on the discussion (see Table 1 and sections 4.2 and 4.3)."

L. 149-152: I understand this choice but would be good to still show the plots versus depth as it is a more common way. Perhaps in supplementary information. Or show plots of salinity versus depth.

We add the Figure 2 of Salinity and temperature against depth. We also add the superimposed casts versus depth in Appendix 2 and some examples of individual cast versus depth in Appendix 3

L. 152: "the processes" – which ones? We now only stipulate that salinity follow better the water masses

L. 152: "overall much lower salinity". Maximum values declined from ~20 to ~16, I wouldn't call that 'much lower' we deleted "much"

L. 155: atmospheric pCO2 was likely higher than 400 uatm. We change this value by 407 uatm (Chen at al. 2020)

L. 162: I wouldn't call a zone with an oxygen concentration of less than 1 uM sub-oxic (I am in general not an advocate of this term), but rather anoxic Suboxic is use to differentiate this zone to the sulfidic zone (also anoxic)

Fig. 1: I am not sure how correct it is to use a linear transformation for plotting for a non-linear variable like pH This plot is only informative and does not contribute directly to any calculation, so the linear transformation is only functioning as a zoom.

L. 171-174: I can see this pattern in NO2 in the 2017 data but not really in the 2018 data. Do you have an explanation for this? It is now precised in the first part of the discussion : " the significant nitrite accumulation […] is not visible in 2018 probably because the higher O2

concentration in 2018 accelerate nitrite oxidation into nitrate and prevent any significant accumulation."

L. 175: This reads a bit odd. I think what you mean to say is that any O2 diffusing downwards would react with upwards diffusing Mn2+. corrected

L. 184: "emerging picture" removed

Fig. 2: mistake in caption; this is nitrite, not nitrate corrected

L. 200-207: If I remember correctly (I didn't look it up), in the model used by Cai et al. (2017) their results were explained by combining aerobic respiration with nitrification. Can the authors elaborate on the comparison with this study? In general, to me it seems surprising that no nitrification would take place.

Cai et al 2017 does not invoke nitrification, in surface estuary nitrification is frequently considered negligeable in regards to TA changes (see Abril et al. 2003 for example). Additionally, we never wrote that nitrification does not take place; it is just not enough important to significantly change the TA and DIC signature. However, in the ILO zone in 2018 our initial consideration that only AR occur can be improved by taking into account both CD and Nit, we added:

"Deeper, in the ILO zone, $\Delta TAex/\Delta DICex/\Delta AOU = 0.2/1/1.25$ (Fig. 5a and 5b) results mainly from AR (0.15/1/1) with possible addition of CD and Nit, the exact signature being fitted for 0.54 CD and 0.46 Nit for 1 AR (4$^{th}$ line in Table 2), in close continuity of AR and CD relative rates from the overlaying layer. This important nitrification is also in good agreement with the lack of nitrite build up in the ILO zone and the relatively high oxygen concentration (at 105 µM) in the ILO zone in 2018 which is able to sustain nitrification."

L. 216: "as was observed for other years" – including 2017 or not? The surface sampling was not precise enough for 2017 cruise, but in august 2016 Chen et al. (2020) report pCO2 below 300 µatm at station 858.

L. 219 – 224: Good that saturation calculations were done here, although it would be good to actually present the (range of) values. Is there a logical source of calcite in this part of the Chesapeake Bay in 2017 that would support this hypothesis? (especially given what is written in L. 229 – 230).

Su et al 2021 report $Ca^{2+}$ production for august 2016 indicating that carbonate dissolution can occur in the Chesapeake Bay. The calculation of carbonate saturation shows that undersaturation more pronounced in 2018 in agreement with the fact that the DIC/TA approach identify CD only for 2018. Section 3.1 explains:

"The $Ca^{2+}$ concentrations observed by Su et al. (2021) and during the 2018 cruise (data not shown) vary linearly with salinity (calcium excess stay below 200 µM or 10% of total Ca). Assuming similar behaviour in 2017, calculations show that the whole water column (except 4 samples from the PP zone) is under saturated (0.36<$\Omega$cal<1; mean=0.68) with respect to calcite in 2018, while undersaturation is only valid below S=10 in 2017."

Also, are any analyses done on the type of algae that would contribute to PP? we have not made any characterisation of primary producers.

L. 237: "never reported in the literature" – perhaps in estuaries or using this particular metric or $\Delta TAex/\Delta DICex$ (although I am not even certain about this). But ratios of TA/DIC exceeding 2 have been discussed in earlier works.

We extended the bibliography to calculate $\Delta TAex/\Delta DICex$ in other environment and summarized it in the section 4.4, but ratio above 2.3 were not found in our short review. If the reviewer has the opportunity to send us any reference we could include them in the discussion.

L. 252 – 258: My main issue with this discussion is that there can be more fates of H2S than only discussed here, each having a different $\Delta TA/\Delta DIC$ ratio. I am not sure that the choice of reactions discussed here and given in table 1 is properly substantiated, especially since no actual modelling has been conducted …

We improve the discussion about the candidate reaction selection in the new section 4.3. About the fate of H2S, we include reoxidation into sulphate or into $S_0$ precipitation as FeS or as FeS2… to my knowledge it corresponds to the consensual main species $H_2S$ can change in. We think the Table 1 well describe the possible fate of $H_2S$.

… and since solid S species or MnCO3 have not been measured.

We agree that our publication would have been reinforced by S and MnCO3 measurement on suspended material. However, we didn't initially expect that MnCO3 would have an important role before sampling. Whatever, it would probably not change anything: as discussed, the MnCO3 precipitation does not occur in the water column (TA build up is too important and Rhodocrosite is undersaturated) but in the sediment. The development of anoxia at station 858 corresponds to an upward diffusion of the redox front that with them the pore water TA and DIC signature. We add additional description about the Mn content in the Chesapeake sediment from other publications showing than Mn is very dynamic in the Chesapeake Bay and that important part of MnOx deposited rapidly turn into MnCO3.

As a result, the authors cannot state whether the formation of MnCO3 is actually important in the Chesapeake Bay. In fact, the authors state this to some extent themselves in L. 269-270.

We clarify the writing to not state that MnCO3 is important, but to only state that the model suggests that MnCO3 is important… which is confirm from a sediment point of view by other publication.

L. 279: I don't understand the unit of $\Delta TAex$ here – or is something else meant? This has been removed.

---

## Referee Report (RR1)

Review of paper "**Influences of manganese cycling on alkalinity in the redox stratified water column of Chesapeake Bay** " by Aubin Thibault de Chanvalon, George W. Luther, Emily R. Estes, Jennifer Necker, Bradley M. Tebo, Jianzhong Su, Wei-Jun Cai

The paper deals with a very important topic which is the transport and transformation of material in estuaries and the role of filter (or reactor) that these estuaries provide. Furthermore, the paper deals with one of the largest and the most studied estuaries on Earth (Chesapeake Bay) which is submitted to numerous anthropogenic pressures. Despite the previous papers including the recent ones by Su et al. (2020-2021), the carbonate system, which is the "currency" of all carbon exchanges in the aquatic environment is still poorly known.

Main comment:
In the present paper, Thibault de Chanvallon et al. explore the role of metals (Fe and Mn) and *in situ* transformation in the carbon biogeochemical cycle. They spend a great deal of time (and text) to convince us that *in situ* transformations are happening and that the observed profiles (or pseudo-profiles as they are plotted against salinity and not depth) are due to complex transformations involving precipitation of carbonate (MnCO3), several biogeochemical "suboxic" pathways with Fe and Mn, and some anoxic pathways. The demonstration is convincing (up to a certain point see comments below), but in a short final paragraph they admit that the biogeochemical reactions do not occur in the water column (hence not *in situ*) due to the lack of reagents (MnOx) or undersaturation with respect to MnCO3. According to them, these transformations rather occur in the sediments, and their by-products are then transferred to the water column. This is in complete contradiction with their statements (including the abstract) that the reactions occur in the water column and change its chemical composition. Furthermore, the paper provides no evidence that these reactions are occurring in the sediment (porewater or sediment profiles, incubations, …). The authors should deeply rework their paper in order to include the benthic source of transformation from the start of the paper instead of stating that at the end.

Other Comments:
1- Illustrations: In general, the graphs are of poor quality and very hard to read. Figure 3 and 4 that present the main results of the paper are hard to read as the symbols are too small (and very often quite similar), and the axis should be splitted in multiple axis (a number of software do that very nicely!) in order for the reader to access the data values. Just one example of haow hard it is to read data from the graphs: for pH, quoted in the graph legend of Fig. 3: pH = 7.175+DpH/300!! Hard to recalculate individual pH values without a calculator! Please add multiple axis and change symbol size and shape.
2- Too much generalities: The paper contains a large number of general sections with Figures and equations which are long and probably unnecessary. Especially, section 2.3.1 "Identification of biogeochemical processes…", is too long, verbious and not so clear. It is more textbook matter when presenting mixing models (lines 110 to 120 including equations 1 and 2 and Fig 1). I would consider shortening this part especially regarding the fact that "in situ transformations" are ultimately replaced by "transfer from the sediment".
3- Treatment of error for $\Delta DIC_{ex}$ and $\Delta TA_{ex}$: I understand that the uncertainty on the measurements of TA and DIC is very small (1 permil), and I acknowledge that. But I question the error calculation (and propagation) of $\Delta DIC_{ex}$ and $\Delta TA_{ex}$. As it is written in the paper, these numbers are differences between the measured values (assume an infinitely small uncertainty) and the mixing curve defined by the end members. The authors quote an uncertainty on the mixing slope of 5% (see also Su et al., 2021). Hence the uncertainty on the difference of concentration (DICobs-DICmix) used for calculating $\Delta DIC_{ex}$ and $\Delta TA_{ex}$ would also be 5% of the DIC or TA at the salinity of the water mass (i.e. about 0.05*2000µM = 100µM). This rapid calculation shows that the error on $\Delta DIC_{ex}$ and $\Delta TA_{ex}$ could be very large

compared to reported values (100-300μM Fig. 3). The authors should spend more time to convince the reader that uncertainties are smaller than my simple calculation or that the observed patterns are statistically solid.

4- Negative biogeochemical pathways: in several occasions (Table 2, line 1 and line 290 "primary production (-AR; Aerobic Respiration)"; line 365 "negative SR; Sulfate Reduction"), the authors provide shortcuts in biogeochemical reactions which are clearly wrong. Primary production process is definitely not the negative aerobic respiration except in some mass balance equation summarizing the effect of these processes on water chemistry. Same for sulphate reduction and sulphide oxidation. The biological organisms that conduct these transformations are different, the biochemical pathways are different. The authors should reconsider their way of presenting these biogeochemical processes.

5- Primary production in flood conditions: I think that the mass balance reaction and chemical ratio reaches its limits when the authors propose that primary production occurs during (or right after) the flood in 2018, and is counterbalanced by carbonate dissolution (line 306). It is known that turbid waters during floods prevent primary production because of light shading, and that primary production favours carbonate precipitation due to the removal of CO2 and the increase of pH. Hence, even if the combination of CD-AR (Table 2, line 2) has the right chemical ratio ($\infty$/0/-$\infty$, line 306), it is very unlikely that these processes can occur in the turbid estuarine waters.

6- Line 373: The ratio of chemical elements observed in the sulfidic region are compatible with reactions involving MnOx and MnCO3, yet the ratio of $\Delta DICex/\Delta H2S$ in 2017 and 2018 are not shown in a Figure to ascertain this point. It could be added in Fig.5 on a fourth panel or in another Figure.

7- Line 365: the authors declare that several combinations of reactions (5-6-7 of Table 2) may provide the identified $\Delta Taex/\Delta DICex/\Delta H2S$ of 2.4/1/0 in the suboxic zone. They state that it is not possible to choose between these three reactions based on the above ratio of elements. One possible way to decipher between these combined pathways is the C/N ratio produced by the reaction as they are quite different for reaction 7 than for reaction 5 and 6. The authors should investigate that point.

---

## Author Response (AR2)

Review #2.1

The paper deals with a very important topic which is the transport and transformation of material in estuaries and the role of filter (or reactor) that these estuaries provide. Furthermore, the paper deals with one of the largest and the most studied estuaries on Earth (Chesapeake Bay) which is submitted to numerous anthropogenic pressures. Despite the previous papers including the recent ones by Su et al. (2020-2021), the carbonate system, which is the "currency" of all carbon exchanges in the aquatic environment is still poorly known.

Main comment: In the present paper, Thibault de Chanvallon et al. explore the role of metals (Fe and Mn) and in situ transformation in the carbon biogeochemical cycle. They spend a great deal of time (and text) to convince us that in situ transformations are happening and that the observed profiles (or pseudo- profiles as they are plotted against salinity and not depth) are due to complex transformations involving precipitation of carbonate ($MnCO_3$), several biogeochemical "suboxic" pathways with Fe and Mn, and some anoxic pathways. The demonstration is convincing (up to a certain point see comments below), but in a short final paragraph they admit that the biogeochemical reactions do not occur in the water column (hence not in situ) due to the lack of reagents (MnOx) or undersaturation with respect to $MnCO_3$. According to them, these transformations rather occur in the sediments, and their by-products are then transferred to the water column. This is in complete contradiction with their statements (including the abstract) that the reactions occur in the water column and change its chemical composition. Furthermore, the paper provides no evidence that these reactions are occurring in the sediment (porewater or sediment profiles, incubations, ...). The authors should deeply rework their paper in order to include the benthic source of transformation from the start of the paper instead of stating that at the end.

The new version of section 2.4.1, better describes the conditions required for the "reaction driven" interpretation. It is now demonstrated that a straight line in a portion of a TAex VS DICex scatter plot indicates that one local endmember has been previously produced from the other by a chemical reaction of $\alpha_{TAex}/\alpha_{DICex} = \Delta TAex/\Delta DICex$. (l157-162)

"However, in a stratified water column, not only one but several successive reactions occur, requiring many integrations of equation (3). On the boundary of each space portion with constant $\alpha_C/\alpha_D$, specific local endmembers are defined with concentrations at steady state fixed due to the ongoing reactions and not due to the inertia of large body of water. The general case is not straightforward to solve but in the particular case where the C versus D plot represents a straight line in a portion of space, the equation (4), still valid in each portion of space, indicates that G=0, thus that the local endmembers are maintained in steady state by a chemical reaction with a similar stoichiometry that the one that produced them, *i.e.* $\Delta C/\Delta D = \alpha_C / \alpha_D$."

The short final paragraph indicates that the local endmember previously produced moved up as the summer begin has been rewritten. (See additional information in reviewer #2 answer.) Therefore, there is no anymore contradiction between the conclusion and the demonstration. The benthic source is now included from the start of the paper (including abstract).

Other Comments:

1- Illustrations: In general, the graphs are of poor quality and very hard to read. Figure 3 and 4 that present the main results of the paper are hard to read as the symbols are too small (and very often quite similar), and the axis should be splitted in multiple axis (a number of software do that very nicely!) in order for the reader to access the data values. Just one example of how hard it is to read data from the graphs: for pH, quoted in the graph legend of Fig. 3: pH = 7.175+DpH/300!! Hard to recalculate individual pH values without a calculator! Please add multiple axis and change symbol size and shape.

A new version of Figure 3 and 4 is proposed that follow the reviewer's recommendations (see new Figures).

2- Too much generalities: The paper contains a large number of general sections with Figures and equations which are long and probably unnecessary. Especially, section 2.3.1 "Identification of biogeochemical processes...", is too long, verbious and not so clear. It is more textbook matter when presenting mixing models (lines 110 to 120 including equations 1 and 2 and Fig 1). I would consider shortening this part especially regarding the fact that "in situ transformations" are ultimately replaced by "transfer from the sediment".

Section 2.3.1 has been rewritten, including the suppression of ll110-120 and Figures 1 and 2. It has not been reduced in length since additional more detailed argumentation is now provided.

3- Treatment of error for $\Delta DICex$ and $\Delta TAex$: I understand that the uncertainty on the measurements of TA and DIC is very small (1 permil), and I acknowledge that. But I question the error calculation (and propagation) of $\Delta DICex$ and $\Delta TAex$. As it is written in the paper, these numbers are differences between the measured values (assume an infinitely small uncertainty) and the mixing curve defined by the end members. The authors quote an uncertainty on the mixing slope of 5% (see also Su et al., 2021). Hence the uncertainty on the difference of concentration (DICobs-DICmix) used for calculating $\Delta DICex$ and $\Delta TAex$ would also be 5% of the DIC or TA at the salinity of the water mass (i.e. about 0.05*2000µM = 100µM). This rapid calculation shows that the error on $\Delta DICex$ and $\Delta TAex$ could be very large compared to reported values (100-300µM Fig. 3). The authors should spend more time to convince the reader that uncertainties are smaller than my simple calculation or that the observed patterns are statistically solid.

The reviewer calculates the uncertainty in a similar way the $\Delta DICex$ is calculated between x1 and x2 ($\delta(x)$ being the uncertainty on x), i.e.

$\delta (\Delta DICex) = \delta (DICex (x=x2) - DICex (x=x1))$

$\delta (\Delta DICex) = \delta (DICobs (x=x2) - DICmix (x=x2) – DICobs (x=x1) + DICmix (x=x1))$

$\delta (\Delta DICex) = \delta (DICobs (x=x2)) + \delta (DICmix (x=x2)) + \delta (DICobs (x=x1)) + \delta (DICmix (x=x1))$

With $\delta(DICobs) \sim 0$ and $\delta(DICmix) \sim 100$ µM it cames

$\delta (\Delta DICex) \sim 200$ µM

thus $\delta (\Delta DICex)/\Delta DICex \sim 100\%$ !!

However, this approach does not take into account the fact that the error on the slope would be the same for DICmix (x=x2) and DICmix(x=x1) and the difference between these two values cancel most of the uncertainty associated to the slope of the mixing line. The reason is that the difference of salinity between x1 and x2 is much lower than the sum of the salinity of x1 and x2. Thus, we have much less uncertainty on

ΔDICex (that is about 5% see below) than on DICex (that is about 50%). It can be demonstrated by considering the relation
ΔDICex = ΔDIC – sml_DIC x ΔS, there we have
δ (ΔDICex) = δ (DICobs (x=x2)) + δ (DICobs (x=x1)) + δ (sml_DIC x ΔS)
With δ(DICobs) ~0 it cames
δ (ΔDICex)/ΔDICex= δ (sml_DIC)/sml_DIC + δ (ΔS)/ ΔS
δ (ΔDICex)/ΔDICex= δ (sml_DIC)/sml_DIC=5%

We add a sentence to clarify this point to the reader (l154-156)
Posing δ(x) as the uncertainty on x, we get equation (6) that describes the fact that the uncertainty is much lower on ☐DICex than on DICex because most the error associated with the calculation of the endmember is cancelled when calculating the difference of DICex on two points with close salinity:

4- Negative biogeochemical pathways: in several occasions (Table 2, line 1 and line 290 "primary production (-AR; Aerobic Respiration)"; line 365 "negative SR; Sulfate Reduction"), the authors provide shortcuts in biogeochemical reactions which are clearly wrong. Primary production process is definitely not the negative aerobic respiration except in some mass balance equation summarizing the effect of these processes on water chemistry. Same for sulphate reduction and sulphide oxidation. The biological organisms that conduct these transformations are different, the biochemical pathways are different. The authors should reconsider their way of presenting these biogeochemical processes.
We agree that the organisms' involved and biogeochemical pathway differs between forward and backward overall reactions. We add this precision to prevent any misunderstanding for the reader. Presentation of primary production is now (l.300):
"primary production (whose overall mass balance equation is here summarized as negative AR)"
And lines 381-383:
"Combinations without MnR-MnC, however, lead to a negative SR whose overall equation could be interpreted as a possible small participation of anoxygenic phototrophic (purple) bacteria (Findlay et al., 2015, 2017) but are not considered further as the amount of ΔTAex involved would be tiny."

5- Primary production in flood conditions: I think that the mass balance reaction and chemical ratio reaches its limits when the authors propose that primary production occurs during (or right after) the flood in 2018, and is counterbalanced by carbonate dissolution (line 306). It is known that turbid waters during floods prevent primary production because of light shading, and that primary production favours carbonate precipitation due to the removal of CO2 and the increase of pH. Hence, even if the combination of CD-AR (Table 2, line 2) has the right chemical ratio (∞/0/-∞, line 306), it is very unlikely that these processes can occur in the turbid estuarine waters.
We recognise that simultaneous primary production with carbonate dissolution is counterintuitive since more frequent combination such as aerobic respiration with carbonate dissolution is more common and are also observed here (table2, line3). However, we measure an increase of oxygen and an increase of alkalinity with no DIC changes… which is also uncommon. Note that in the upper Chesapeake Bay, water pCO2 is naturally high and is undersaturated with respect to CaCO3 and that biological production did occur in areas of low turbulence and sufficient light penetration. Besides, an increase of $Ca^{2+}$ is also observed simultaneously. We think

that the original manuscript takes enough caution to describe this feature (l. 318-321), and does not extend on this sensitive point:

"This original signature can be modelled by the combination of simultaneous carbonate dissolution (CD), the water column being undersaturated, and PP, in equal proportion (2nd line in Table 2); the carbonate dissolution buffers the DIC consumption by the PP."

6- Line 373: The ratio of chemical elements observed in the sulfidic region are compatible with reactions involving MnOx and MnCO3, yet the ratio of $\Delta DICex/\Delta H2S$ in 2017 and 2018 are not shown in a Figure to ascertain this point. It could be added in Fig.5 on a fourth panel or in another Figure.
We agree with the reviewer and add an H2S versus TA plot in the Figure 5d

7- Line 365: the authors declare that several combinations of reactions (5-6-7 of Table 2) may provide the identified $\Delta Taex/\Delta DICex/\Delta H2S$ of 2.4/1/0 in the suboxic zone. They state that it is not possible to choose between these three reactions based on the above ratio of elements. One possible way to decipher between these combined pathways is the C/N ratio produced by the reaction as they are quite different for reaction 7 than for reaction 5 and 6. The authors should investigate that point.
Unfortunately, we did not measure the C/N ratio in organic matter during this campaign.

Review #2.2

I am reviewing this manuscript for the second time and feel that the authors have tried to incorporate the comments of both earlier reviews. The split into results and discussion helped to improve the manuscript and section 2.3.1 is a nice addition (but: see below). I feel that I am less critical on the manuscript than last time, but at the same time I am not yet on the point where I can recommend publication.

General comments
The introduction lacks a proper build-up. In the new section on anoxic environments, suddenly alkalinity is used without introducing its link to carbonate dynamics in the first paragraph. I also don't understand the use and discussion of 'charge transfer'; for example, when Fe and Mn oxides are used for OM decomposition, they still change alkalinity. Finally, this section contains quite some inaccurate formulations: e.g., metal oxides are not a pathway (L. 42) and are not always transformed into sulphur or carbonate species; they can also remain in dissolved form.
We agree with the reviewer and rewrite in depth the second section of the introduction

Section 2.3.1 is a nice addition to the manuscript but not easy to read. I like to think that I am mathematically inclined, but I still don't follow all the reasoning here. First, the explanation now uses a mixture of 'hypothetical species' C and D in the equations, and TAex and DICex as examples in the text. However, for example C and D in Eq. 3 to 5 cannot directly be replaced with TAex and DICex, as they cannot be part of a reaction equation. So I suggest that you clarify how C and D are linked to TAex and DICex.
We change significantly the section 2.3.1 to answer to both reviewers' comments. In particular it is now precised that (l124-125)
"Equation (3) is also true for any linear combination of a solute with a conservative element such as the salinity, S, because $\alpha_S = 0$, hence:
$$\frac{d^2C + kS}{dx^2} = \frac{d^2C}{dx^2} + k\,\frac{d^2S}{dx^2} = \frac{d^2C}{dx^2}$$
and later (l.137-139)
"In this study, while $\Delta TA \neq \Delta DIC \neq 0$, the excess of TA (TAex) and the excess of DIC (DICex) are calculated by linear combination with salinity to be equal to zero for the upstream and downstream endmembers reaching the condition $\Delta TAex = \Delta DICex = 0$."

Second, I don't understand where the 0.1 in Eq. 6 is coming from.
The 0.1 in Eq (6) come from the addition of two uncertainties of 5% as precised in l 141-143:
"The oceanic endmember varies mainly with season (Cai et al., 2020) and a maximal change of 50 µM results in 5% uncertainty on the slope of the mixing line."

Third, I don't think that the bow and the set of spears are as different as you present them. In my view, the bow is simply representing how the relative weight of the various reactions that make up each spear may change as a function of salinity.
These expressions and the related explanation have been deleted in the new version

At the same time, each of the spears can still include an additional source (e.g. from sediments or lateral exchange) and you seem to ignore this possibility in your calculation of v and vi in Eq. 9 (L.167). When looking back at the comments on the earlier version, I see that this has also been pointed out then.

It is now clearly precise that the absence of a third endmember, or of significant lateral mixing is necessary to apply the "reaction driven" approach (l163-166)
"Therefore, in a system defined between only two endmembers, away from atmospheric exchanges, in case of turbulent diffusion mixing, at steady-state and with negligible lateral mixing, the "reaction driven" approximation allows us to interpret linear variations of TAex versus DICex as a sum of biogeochemical reactions spread all over the water column that can be broken into several discrete reaction zones."

The final part of the discussion becomes very confusing (L. 404-419). How can you use a reaction driven approach if you explicitly state here that external inputs from the sediments are required? This seems very contradictory. It is only possible if 1) you include sediments as part of your system, which doesn't seem to be the case; and 2) extend the timescale, but then the steady-state assumption doesn't hold anymore. This point really needs clarification, and in fact probably means that the distinction between bow and set of spears cannot be drawn as black-and-white as this manuscript does.

We agree with both reviewer that this point needed to be clarified. Section 2.3.1 has been rewritten to fine-tune the demonstration introducing the concept of *local endmember* to explain steep changes of DIC/TA slope (see answer to reviewer #1). We also tried to fluidize the explanation in the first section of 4.5 (l422-430):
This mass budget discrepancy cannot be solved invoking suspended material since the 88 µM of $MnO_2$ would require a suspended material concentration of about 4.4 g $L^{-1}$ (assuming an average concentration of 20 µmol $g^{-1}$ of Mn), which is again one or two orders of magnitude higher than the 0.01 – 0.1 g $L^{-1}$ usually found in the Chesapeake Bay (Cerco et al., 2013). However, a fast settling rate could satisfy and explain the discrepancy between water and solid concentration. But another process dephasing aqueous from solid reaction products is also possible at station 858, since the dissolved phase could have moved up, rather than the particles settling down. In this case, the SR-SMnC reaction was not happening only in the water column of the Chesapeake Bay and part of the TAex and DICex pool could have been produced in the sediment during the previous year, then diffused out of the sediment simultaneously with other reduced elements as the summer begins.
A specific section has also been written (l444-452):
The assumptions detailed in section 4.1 permit the "reaction driven" approach to be reconsidered in regard to this sediment efflux scenario. Indeed, the sediment efflux does not need to be considered as an additional endmember, that would violate the third assumption, since its salinity and the pore water concentrations results from the upstream and oceanic endmembers superimposed to geochemical reactions. In section 2.3.1, we point out that equation (4) was valid in between each local endmembers and that the straight lines on the ΔTAex/ΔDICex plot between them indicates that the local endmembers are maintained in steady state by a chemical reaction with a similar stoichiometry. These results indicate that even if most of the $MnCO_3$ was produced when the local endmembers were localised in the sediment, a-their migration does not alter the TAex/DICex signature and b- the chemical reaction

that produced them is still ongoing at sufficient rate to maintain a steady state characterised by the steep changes of direction observed in Fig. 4a.

Detailed comments per line

L. 75–92: This section is very detailed in its experimental description but it lacks clarity on which Mn and Fe species are actually measured. I was only able to deduce this information from the results.
We add these information (l. 84-85):
Iron was measured on both bulk and filtered samples using the ferrozine method (Stookey, 1970): after HCl acidification (for Fe(II)) and an optional reduction step (for Fe(III)+Fe(II))

L. 102-103: I found this difference between the 2017 and 2018 campaigns quite interesting. Can you link it to the higher inflow of 2018 making e.g. the used equilibrium constants less reliable?
In some point yes. I think the relation is mostly conduct by nitrate and nitrite concentration: a) in 2017, the non-carbonated-TA find its minimum at the maximum of nitrite and b) the higher $NO_2^-$ concentration in 2017 correspond to the lower non-carbonated-TA. The $NO_3^-$ dataset is lacking, but the higher runoff in 2018 could have diluted the fertilizers…

L. 110-112: This is a very complex sentence. Please try to simplify.
L. 115-116: Also Eq. 1 is not valuable in case of such a change.
Figure 1 caption: not sure what you mean by 'a segment' here.
L. 126: Why not visually add this excess to Fig 1 in order to try to link C and D better with TAex and DICex?
The section of concern for the 4 previous comments has been removed

L. 130: "has to be determined" I would say that this is your choice and I would be really curious to know how much uM this TA endmember would have to change in order to add 5% uncertainty on the slope of the mixing line (as you discuss for the oceanic endmember).
We add the sentence (l. 145-146)
Between the two campaigns, the upstream endmember changed by 77 µM generating 5% of change on the slope (see Fig. A1).

 Your previous manuscript version had a big discussion on the upstream endmember and although I understand your current choice, I would like to substantiate a bit more that it is 'less sensitive to short-term changes' (L. 133)
We precised the sentence (l 148-150):
However, it corresponds to a larger water mass pool, less sensitive to short term changes, with a residence time being higher than 240 days in the Chesapeake upstream part (Du and Shen, 2016), and thus is more likely to satisfy the condition of stability of the endmember, which is a prerequisite of the steady state assumption.

L. 140: Here you should use "net stoichiometry" (or "apparent stoichiometry" as you use later in this section), it is still a mixture of several reactions.

We add the adjective

L. 142-143: I'm not an expert in error propagation, but shouldn't the uncertainty be equal to the square root of the sum of squared uncertainties?
The square root of the sum of squared uncertainties is often used for analytical error propagation but will be inferior of the sum of uncertainties we are using. So, for sake of simplicity, we prefer to keep the equation unchanged

L. 147-149: I don't understand this sentence, Eq.6 doesn't describe a single solute. Shouldn't this refer to Eq. 4 instead?
True, we change the equation

L. 150: I don't understand this steady state assumption in the context of what you state in L.146-147 on the temporal evolution of water masses. Is this because you exclude additional sources from your model? (See comment above on spears versus bow)
This sentence has been removed in the new version of the section 2.3.1

L. 177: I feel that Eq. 12 needs a bit more credit to the earlier approaches linking charge with alkalinity than a short mention in L. 186-187 alone.
We add the equation 32 from Wolf-Gladrow et al. 2007 for comparison (l.195-199):
Eq. (10) is equivalent to those published in Soetaert et al. (2007) or Wolf-Gladrow et al. (2007) whose equation 32 can be refined considering that :

$$\sum_i z_i^{pH=4.5} = [Na^+] + 2\,[Mg^{2+}] + 2\,[Ca^{2+}] + [K^+] + 2[Sr^{2+}] + \cdots$$

$$- [Cl^-] - [Br^-] - [NO_3^-] - \cdots TPO4 + TNH3 - 2TSO4$$
$$- THF - THNO2 - \cdots$$

However, Eq. (10) is more general. For example, in suboxic water, specific species such as polysulfides (as $HS_8^{2-}$, Rickard and Luther, 2007) and in highly productive environments, carboxylic groups from DOC can be easily added as soon as the bearing charges at pH = 4.5 are known.

L. 196: Similar zonation yes, but the 2018 profile appears noisier and the transition somewhat shallower (at 5-6 m depth rather than ~7m depth in 2017)
Yes, it is also one of the reason why plot against salinity is preferred farther.

L. 199-203: This is again a very complex sentence. You write about pCO2 lower than atmospheric but then mention values of 505 and 770 uatm? I don't follow this.
We clarify this sentence, indeed the average on the zone considered are above atmospheric value but some sample show lower pCO2 (l. 209-214).
Below, at 3 m depth, a subsurface layer (named primary production zone or PP in Fig. 2) is characterized by a high amount of $O_2$ (about or above 100% saturation), high pH (about 8; 8.11 ± 0.07, n=13 in 2017 and 7.94 ± 0.08, n=14 in 2018) and high day to day temperature variation (above 1 °C between different days). The layer presents relatively low $pCO_2$ (505 ± 75 µatm, n=13 in 2017 and 770 ± 130 µatm, n=14 in 2018) with minimal values at 110 µatm in 2017 and 205 µatm in 2018, which are below the atmospheric $pCO_2$ of 407 µatm (Chen et al., 2020)).

L. 212: I don't like the term 'suboxic' anyway (I prefer hypoxic; in fact your ILO zone can be called hypoxic zone) but <1 uM already is anoxic.
We didn't call "hypoxic" the ILO zone as some scientists will not like that an hypoxic zone with 105 µM of O2.

Anoxic and euxinic/sulphidic have clear different meanings (anoxic meaning without oxygen, euxinic meaning free sulphide present), so using anoxic here is more correct. Then, in L. 223, you can write "the transition from sulphidic to anoxic zone" which also seems more correct.
Anoxic and euxinic/sulphidic have clear different meanings but a sulphidic water can be anoxic too, which can lead to misunderstanding. By the way, some sulphidic water can have also oxygen since reduction of oxygen by free sulphide is rather slow… we add the precision l.223-224 and change the zone name from suboxic to anoxic:
Deeper, where the oxygen is not detectable (< ~ 1 µM) and in absence of free sulphide, the so-called anoxic zone ...

L.241: I'm not sure if you can deduce from Fig 3 and 4 that the stratification is similar. In fact that is more clearly shown in Fig. 1. Maybe a different wording would better fit what you want to describe here (e.g. zonation?)
We change "stratification" by "zonation"

L. 249: A zone with neither oxygen nor sulphide present is an anoxic zone, not a suboxic zone. See my earlier comment on this topic.

L. 264-266: As said before, I don't think this is "either/or". It might be more valid to say that one dominates the other (i.e. reactions dominate over mixing, in this case).
We suppress the sentence, and added the concept of domination in the following sentence (l.275-278)
At station 858, the steep gradient observed, for example the pH and $pCO_2$ gradients in the PP zone, the $O_2$ and $NO_2^-$ gradients above the anoxic zone and the Mn, Fe and $H_2S$ gradients at depth, suggest that ongoing in situ processes control the changes of concentrations and dominate the time-dependent endmember variability or the mixing with an unknown third endmember.

L. 274-278: This is quite a list of assumptions – good that they are explicitly mentioned. I don't understand the difference between #2 and #3 – what do you mean with 'starting point'? If that is in time, it is similar to a steady state, isn't it?
We agree with the reviewer and modify this sentence as (l.283-287):
Assuming 1) that mixing is efficiently described by vertical turbulent diffusion mixing, 2) that the measured concentrations correspond to a steady state – no changes observed over the 1 week sampling, 3) that no additional endmember contributes significantly to the excess calculation, in particular that the samples are isolated from atmospheric exchanges and 4) that lateral mixing is negligible, which is equivalent to the lateral invariance of the system – as in the stratified water column of station 858;

L. 282: "this interpretation does not identify reactions with minor impact on the carbonate cycle"- because of a low rate, or their stoichiometry, or both?

Because the product of rate x stoichiometry is too low compared to other reactions. We add a reference to the equation (7)

$$\alpha_C = \frac{1}{v} \sum_i \alpha_C^i v^i$$

L. 292: "which corresponds to the occurrence of only net aerobic respiration (AR)" – two comments: 1) add this indicates that AR > -AR (you use the same symbol for AR and net AR now),
we add the precision (l.304-305)
Note a - that "net aerobic respiration" indicates that primary production is possible at a significant rate, but slower than AR;

and 2) given that the slope is 0.1 and ΔTA of AR is 0.15, something else must have occurred with alkalinity as well, unless you have clear indications that OM was very different from Redfield ratio. If you were to fit ΔTA rather than ΔAOU, you would probably have around 15-20% of the produced NH4+ nitrified, I guess? What would then be the resulting ΔAOU? Are there indications that it's more appropriate to fit ΔAOU rather than ΔTA?
Since AR only fit rather well the dataset (Occam's razor), we didn't investigate further. However, we recognise that the fit to ΔTA gives very interesting value. We modify the Table 2 accordingly and add the sentence (l.307-308)
A combination of 1 AR and 0.025 Nit (nitrification of 16% of the produced $NH_3$) improves the modelled value to 0.1/1/1.05.

L. 306: I don't understand the infinity symbols here. Yes, ΔDICex = 0 but since you compare the slopes of three different species, ΔTAex/ΔAOU will not equal infinity. Otherwise, you have to present ΔTAex/ΔDICex/ΔAOU differently and make it clear that you always compare ΔTAex/ΔDICex and ΔAOU/ΔDICex, as you do in Fig 5 and Table 2. But from the way it is in the text, and also because you fit three reactions to three equations, this isn't obvious at all. The same applies to the presentation of ΔTAex/ΔDICex/ΔH2S later on.
The signature has been calculated and added in the text (l.317-321).
In Fig. 4a and 4b, this process translates into a vertical distribution at DICex = 40 μM with ΔTAex/ΔDICex/ΔAOU = 1.37/0/-1. This original signature can be modelled by the combination of simultaneous carbonate dissolution (CD), the water column being undersaturated, and PP, in equal proportion (2nd line in Table 2); the carbonate dissolution buffers the DIC consumption by the PP. Note that the ratio between ΔTAex/ΔAOU implicates significant nitrification.
And later (l.334-335)
Additionally, in 2017 the ΔTAex/ΔDICex/ΔAOU system indicates weak nitrification, while in 2018 significantt nitrification in the ILO and PP zones are suggested by the "reaction driven" approximation.

L. 314-322: Why would nitrification occur in this zone in 2018, but not in 2017 or in the zone above? The reasons regarding kinetics (L.296) prevail here as well. Are there logical reasons to assume that kinetics are limiting above (where O2 is higher), but not in this zone?
We add an addition hypothesis concerning this point l.335-338
The role of nitrification in explaining TAex depletion is only hypothetical since no direct measurement of $NH_4^+$ and $NO_3^-$ were not performed. In particular, TAex

depletion is particularly intense during high flow, high suspended particles season and could be produced by $NH_4^+$ adsorption to the particles rather than by nitrification.

L. 327-329: Well, that depends on what you want to know. I am not sure if I agree; it depends on whether the combined $\Delta TAex/\Delta DICex/\Delta H2S$ can be derived by multiple combinations of multiple processes.
We precise "to fit with the "reaction driven" interpretation"

L. 336-344: reading this makes the focus on charge in the introduction much more understandable. I would move this text to the introduction and merge with the current paragraph (still taking into accounts the comments).
The concepts of "charge transfert" has been removed from the introduction

L. 345-347: But you measured these species, didn't you? Why don't you make this decision based on your measurements, such that you can substantiate this choice? When looking at your measurements, I am not sure if your measurements substantiate this choice; especially given that you discuss their dynamics in the result section as well.
We add the precision (l.363-365)
These species are usually recycled rapidly and hold a role of catalyser or electron shuttle between other redox species and did not reach 10 µM during the campaigns (Fig. 3).

L. 351-353: I don't know what you want to achieve by including this reason, but the fact that you cannot distinguish SR followed by H2S oxidation from AR in your model, does not mean at all that this set of processes isn't important. It just means that you cannot conclude it from your model.
Yes, we add the precision (l. 370-372)
As an example, the chemical equation of SR followed by $H_2S$ oxidation with oxygen is equal to the equation of aerobic respiration: the proposed model confounds both pathways because the resulting chemical changes are similar.

L. 359: "as the only Fe product is FeS or FeS2"- where? In your model or in reality? (i.e. as can be deduced from your measurements)
We now precise (l.377-379)
Direct respiration of FeOOH is also taken into account, but as the only final Fe product in the model is FeS or $FeS_2$, it has to be accompanied by some SR (FeSR-FeS).

L. 363-368: I find this section much more strongly formulated than L.360-362 which, in itself, leads to speculation. So I would revert the order: any reaction in combination with MnR-MnC leads to the production of the ratio, and some of them are more likely than others.
We follow the reviewer recommendation and supress the lines 360-362

L. 370-372: Measured concentration of MnOx are quite low; are they high enough to support this statement?
This point is investigated in section 4.5

L. 372-374: This is quite short; which set is the most likely? Do you expect this to be

the same set as in the anoxic zone? It the same set that in the anoxic zone, but part of the H2S produced has not been oxidized yet.

And what about sedimentary input? (which you discuss earlier that it must be an important source) Again this comes back to the lack of external inputs into your reaction-driven model.
We clarify this point rewriting these sentences (l. 389-394)
Deeper, the vertical gradient of sulphide suggests that part of the $H_2S$ came by diffusion from the sediment's porewater (Fig. 2). The assumptions required for the "reaction driven" approximation are still valid as soon as steady state is maintained by ongoing reactions, even if one of the local endmembers has not been sampled since it is probably located in the sediment. In the presence of sulphide, the $\Delta TAex/\Delta DICex/\Delta H_2S$ signature is 2.4/1/1.2 in 2017 and 2.4/1/3.2 in 2018 (Fig. 4a and 4d) and can be explained by the same combination of reactions without complete oxidation of $H_2S$ from SR to take into account the build-up of $H_2S$ (Table 2).

Table 3: Aren't there studies from the Baltic Sea water column that you could include here? That system may be more similar to the Chesapeake Bay water column than many of the other systems discussed in this table.
Most of the Baltic Sea water column dataset we found was
Kulinski et al., 2014 whose dataset come from surface water with no indication about the oxygen concentration
And Beldowski et al. 2010 whose dataset encompass both oxic, anoxic and sulphidic water column. However, data was not available (at least rapidly) and would require an extensive data processing to identify the different endmembers (up to 5 are defined in the publication) and calculate the corresponding excess of TAex and DICex (DICex has been already calculated) which is beyond the scope of our study.

L. 398: "The rhodocrosite saturation (Luo and Millero, 2003) is always below 0.3" – where? In the Chesapeake Bay?
We add the precision "in our samples"

L. 399-401: So this boils down to my earlier comment – a reality check on your model results.
L. 404-419: Here I get really confused – see general comment.
L. 420-426: This upscaling seems a bit out of place, given that the Chesapeake Bay may not be representative at the global scale. This is in fact the main conclusion of your study, that you show the exceptional $\Delta TAex/\Delta DICex$ ratio. I would therefore suggest to remove it. We follow reviewers' comment and remove this section

L. 436-439: These lines seem out of place in the conclusions, also in the context of my previous comment. We write again the conclusion, removing the reference to charge transfer concept.

Figure A3: What is meant with the "dMnT…" comments for Cast #10 of 2017? We modify the figure to make it clearer

Technical comments
General: especially the newly written sections contain many typos and sloppy writing.

I did not identify all occurrences but highlighted a few below.

L. 12 (and many more occurrences): use 'dynamics', not 'dynamic'

L. 12: 'carbonate minerals'

L. 39: 'anoxic environments' (reactions are anaerobic)

L. 56: 'a single station' sounds better in my opinion

L. 168: "reaction stoichiometry"

L. 175: "species"

L. 198: Explicitly refer to Fig. 3 here.

L. 209-215: add (n=xx) in between brackets for clarity. Also this text is complex to read.

We gratefully thanks the reviewer for these typos correction that have all been taken into account

---

## Author Response (AR3)

Review #2.1
2nd Review of paper "Influences of manganese cycling on alkalinity in the redox stratified water column of Chesapeake Bay " by Aubin Thibault de Chanvalon, George W. Luther, Emily R. Estes, Jennifer Necker, Bradley M. Tebo, Jianzhong Su, Wei-Jun Cai

The paper has evolved in a positive way since the previous version: better Figures, fair explanation of the error treatment on DICex and TAex, improvement on the "negative" biogeochemical pathways. But they fail to properly answer the other points that I raised (see below my previous comments):
- The relation with the sediment is still not properly documented (any older study at that site or nearby?) but, at least, the point is tackled in the abstract and a paragraph is written to explain this link. The authors should provide literature data concerning the sediment processes.
We added the reference of Aller 2014 (Sedimentary Diagenesis, Depositional Environments, and Benthic Fluxes, in: Treatise on Geochemistry (Second Edition)) for generalities about sedimentary diagenesis reactions. Previous studies of Sholkovitz et al, 1992 and Lenstra et al., 2021 are quoted for local specificities near or the same station.

- The general statements: I recommended on my first review to shorten part 2.3.1. It was changed and some textbook part including the bow and spear section and Figures were removed. But I still find this part unnecessary long and I am convinced that it could be shortened or part of it diverted in an annex.
We follow the reviewer comment by diverted most of the equation in an Annex

- Furthermore, I had troubles during my reading with part 2.3.2. I find the definition adopted for TA (the sum of all charges that each species would have at pH=4.5) not obvious and certainly less useful than identifying bases in solution at the in situ pH and defining which processes consume or produce them. I find this part 2.3.2 not necessary, not well-named ("TA changes indicated by reaction stoichiometry"), and not clear enough to add value to the paper. I would advice to remove it.
The charge approach used to calculate TA, has been developed in previous papers (e.g. Soetaert et al. 2007) to infer the TA changes produced by a reaction that modifies simultaneously the amount of bases in solution and the in situ pH. In case of such a reaction (for example carbonate dissolution) it is not possible to identify rapidly the changes of pH (most of the DIC from carbonate dissolution is going to be $HCO_3^-$, but a small amount will be $CO_2$ and get volatilised, increasing the pH while another amount is going to form $CO_3^{2-}$ and will decrease the pH making the direct calculation of TA very uncomfortable). In contrast, the charge approach indicate that the change of TA will be 2 (for $Ca^{2+}$, DIC does not contribute as it is $H_2CO_3$ -no charges- at pH 4.5). We modify the title and the first sentence of this section to trigger the importance of the charge transfer approach (l.174-175):
"The simplest way to calculate the TA changes induced by an individual reaction is to look at charge transfer induced by the stoichiometry of the given reaction."

- Last point: the possibility of primary production in turbid waters especially during the flood. The authors did not answer to that specific question nor provide data proving that the water was clear enough to allow primary production.
Unfortunately, we did not measure the concentration of suspended particles.

However, we quote Cerco et al, 2013 (l.424); that reports observed values always below 50 mg L$^{-1}$ at our station, and modelled a median below 10 mg L$^{-1}$ even at high runoff.

I think that the paper is heading in the right direction and that the authors should consider the comments above before being published.

Detailed comments (some):
- title: "Influences" should be replaced by "Influence" We change the title accordingly
- line 134 and after: equation numbering is missing, please check!
Equation that are not quoted in the main text were not numbered
- line 165: "spread all over the water column". Add "and also in local sediments"
As described in the lines 150-158, the "zones" (we renamed it "stratum") does not include the sediment, but include local endmembers that could have been produced in the sediment before their migration upward (and before the steady state achievement).
- paragraph 3.2: too many numbers in text, add a Table with all these numbers
All these numbers are extracted from the Figure 3, therefore adding a Table seems unnecessary.
- line 314: provide information on water turbidity to ensure primary production conditions were present at that time (2018)
We change the sentence into (l.312-315):
This original signature can be modelled by the combination of simultaneous carbonate dissolution (CD), the water column being undersaturated, and PP, no important turbidity was visible as modelled by Cerco et al. (2013), in equal proportion (2$^{nd}$ line in Table 2)
-line 380: "Figure 4c demonstrates…" I think it is rather Table 2. Please change! It corresponds to a graphical demonstration since a combination of any arrow in Figure 4c with the arrow corresponding to MnR-MnC can produce the slope of ΔTAex/ΔDICex = 2.4.

Review #2.2

This is the third time I am reviewing the manuscript and I have thus focused on the parts of the manuscript I was most critical about in the previous rounds of review: introduction, section 2.3.1 and discussion, especially the final part. In my view, the introduction has sufficiently improved but the authors still have some work to do on the discussion and especially section 2.3.1.

General comments
The introduction has surely improved compared to the last version. There is now a link between anoxia and the carbonate system, but the unknowns and objective of this study could still be better introduced. Why is it necessary to better constrain the carbonate cycle in temperate microtidal estuaries?

We added a sentence summarizing the issues and limiting knowledge at the beginning of the last section of the introduction (l.53-56) :

"While TA controls the $CO_2$ buffering capacity of the ocean, riverine input of carbonate to the ocean is poorly constrained (Middelburg et al., 2020) and only rare publications take into account the estuarine transformations of the carbonate species (e.g. Su et al., 2020a; Abril et al., 2003) furthermore in a context of oxygen depletion (e.g. Abril et al., 2004). To better constrain the carbonate cycle in oxygen depleted estuaries, …"

Section 2.3.1 has been shortened quite a bit (which is a good thing, from my point of view, as the old Figure 1 was probably redundant) and it is good that the meaning of C and D is now explicitly explained. However, the shortening comes at the expense of the readability of the first section. Eq. (1-5) include a lot of terms that are not introduced (such as alpha / stoichiometry and k / rate constant (I assume)) and I wonder how much is really necessary for the manuscript. In the end, the authors would like to show how TAex and DICex can be used in combination with reaction stoichiometries to infer which processes can explain the observed TAex and DICex combination (L.163-166). This means that Eq. (5) and Eq. (7) are the two key equations. I would suggest to either explain L. 118-140 more clearly (and thus expand a bit) or to keep only the essential information in (my preferred option).

We reduced the section 2.3.1 by 6 equations by diverted the most technical parts in annexes.

Maybe the text in L. 283-289 can be rewritten to include in the method section, because this text is non-technical and clearly outlines the assumptions (and confirms that Eq. (5) and (7) are indeed key).

The assumption are now outlines on l. 120-121 and in lines 159-163.

In case of turbulent diffusion mixing (sometimes called eddy diffusion) in only one direction (no lateral input), at steady state, on a portion of space where occurs one chemical reaction, the changes of concentration, C and D, of two species can be described by equation (1) (see Appendix 1 for more details):

Therefore, in a system defined between only two endmembers, away from atmospheric exchanges, in case of turbulent diffusion mixing, at steady-state and with negligible lateral mixing, the "reaction driven" approximation allows us to

interpret linear variations of TAex versus DICex as a sum of biogeochemical reactions spread all over the water column that can be broken into several reactional stratum. In each stratum, if the local ΔTAex/ΔDICex ratio is constant, it corresponds to the apparent stoichiometry of a combination of the biogeochemical reactions occurring in this stratum.

In the previous version, both the other reviewer and I commented on the fact that it is nowhere made explicit that the reactions in Table 1 take place in the sediments and that sedimentary inputs are thus key. The abstract has now properly been rewritten to include that some of the main reactions take place in the sediment, but this needs to be clearer in section 2.3.1 as well. The authors, in their response to reviewer 1, claim that L.157-162 covers this point, but because this text is so technical this key point doesn't come out strongly at all.

We apologise our difficulty to clearly describe the concepts and thanks the reviewer for his/her remarks since it helps us to clarify the condition of application of the reaction driven approximation. We write again the lines 150-158 to better explain the implication of the local endmember definition;

However, in a stratified water column, not only one but several successive reactions occur, limiting the validity of equation (1) to each reactional stratum. The general case is not straightforward to solve, but in the particular case where the C versus D plot represents a straight line between two endmembers with different concentrations, the previous analyse of equation (2) indicates (second case) that one endmember would have been previously generated from the second by a chemical reaction with similar stoichiometry. Thus, the depths corresponding to the straight line define a reactional stratum characterised by a constant αC/αD and delimitated by two local endmembers maintained in steady state by chemical reactions with similar stoichiometry than the one that produced them, i.e. ΔC/ΔD = αC/αD. The local endmembers should have been produced before the steady state achievement, by a reaction of similar stoichiometry but the reaction could have been faster than the observed one or could have occurred in a different place, including in the sediment.

Moreover, L.165 still mentions "spread all over the water column" which is fundamentally incorrect because the "zones" include the sediments. There are more examples of this (e.g. L.278, "in situ"). In short, the authors have to clearly define that the "reaction driven" approximation includes reactions in water column and sediment from the start.

As described in the lines 150-158, the "zones" (we renamed it "stratum") does not include the sediment, but include local endmembers that could have been produced in the sediment before their migration upward (and before the steady state achievement).

This discussion benefits from always comparing ΔDICex with ΔTAex and ΔAOU or ΔH2S. I agree with the majority of changes done in the discussion and also in the final part. However, I do think that the main points of section 4.5 regarding the role of sediments (L. 444-452) need to be moved to / integrated in the method section because (again) the role of the sediments in the approach has to be clear from the start.

The new version of the lines 150-158 takes into account most of the concept initially written only in the lines 444-452, in particular:

The local endmembers should have been produced before the steady state achievement, by a reaction of similar stoichiometry but the reaction could have been faster than the observed one or could have occurred in a different place, including in the sediment.

Detailed comments per line

L. 22: "carbonate signature" --> I suggest to change to "DIC/TA ratio" because this is what you refer to.

We refer to ΔTAex/ΔDICex ratio. It seems important to not confound TA/DIC with ΔTAex/ΔDICex.

L. 24: "especially in river-dominated environments" --> this now reads as if it applies to all river-dominated environments, but you write yourself that the Chesapeake Bay is quite peculiar (see L. 396). Better rephrase: "the critical role of Mn in alkalinity dynamics in the Chesapeake Bay and potentially other river-dominated environments"

We changed the text according to the recommendation

L. 32: "This disequilibrium" --> which one? The DIC increase without concurrent TA increase mentioned two lines earlier? Specify.

We change by "This cationic deficiency"

L. 36: "accounts for 2/3 of buried carbonate" --> where? Shallow waters globally? Please specify. We change the sentence into (l.35-38):

However, in shallow waters, that accounts for 2/3 of global buried carbonate (Smith and Mackenzie, 2016), carbonate precipitation largely predominates over dissolution and other localised processes may constrain carbonate dynamics (Borges et al., 2006; Lohrenz et al., 2010).

L. 41: Do you mean to say that humans migrate to coastal areas because of global warming, or that global warming contributes to eutrophication? We change the sentence into (l. 43):

The global trends of human migration towards littoral areas and global warming favour eutrophication and a decrease in oxygen levels in coastal water

L. 46: "multiplies the possibilities for" --> strange formulation, what about "enhanced the possible build-up of" Thanks for the proposition, we change the sentence accordingly

L. 49 "negatively charge SO42-" --> do you need to mention the charge now since you have removed it from the introduction? We suppressed the reference to the charges.

L. 53: "TA and DIC concentrations" I wonder here if the TA and DIC concentrations or the fluxes matter. S burial (taking place in the sediment) leads to specific pore water TA and DIC concentrations which impact water-column TA and DIC (the subject of this study) via effluxes. We agree with the proposition and change "concentrations" by "effluxes" since it better prepare the reader to the final story of the study.

L. 165: "spread all over the water column" --> and the sediments. See general comment above. We maintain our formulation since we only interpret signals from the water column. Even if the local endmember has been produced in the sediment, the reaction has still to occur in the water column to maintain the local endmember at the steady state.

L. 277: "in-situ processes" --> as opposed to mixing, but including the sediments. Again, that should be made clear here. See above, the reaction has to occur in situ also to maintain the steady state. If not, the ΔDIC/ΔTA would not be a straight line.

L. 289-291: I am not sure if I find this a clear comparison, and perhaps even misleading as water is also moving in the opposite direction. As proposed, we removed the comparison

L. 294-295: "an excess of ΔTAex" --> this becomes confusing. What about: "even higher ΔTAex"? We changed the sentence according to the comment

L. 308: "the weakness of nitrification" --> rephrase to something like: "the relatively slow nitrification" We changed the sentence according to the comment

L. 321: "significant nitrification". How much approximately / at least? We realized the sign of nitrification should be negative, therefore we changed the sentence into (l. 316-318)

Note that the ratio between ΔTAex/ΔAOU implicates an important nitrate assimilation superior or equal to the amount of N required for the PP, as modelized by negative nitrification in Table 2.

L. 364-365: these low concentrations may have been found in the water column, but not in the sediments where you propose that the reactions take place. So can you use it as a reason here? Yes : the $Mn^{2+}$ or $Fe^{2+}$ produced in the sediment have been probably oxidized during the dephasing as the local endmember moved up into the water column. So the fingerprint on TA/DIC include this precipitation of $Mn^{2+}$ and $Fe^{2+}$ and have to be interpreted with solids as product of reaction.

L. 386: "at this site" --> in the sediment or water column? We replace by in this water column

L. 390-392: This is quite an assumption without any sediment measurements, although I agree that on a 10-day period is it likely met.

L. 396: This line exactly indicates why the final sentence of the abstract needs rewriting.

L. 416: Title needs to be changed because global budget is removed. We changed the title into Local budget

Technical comments (I have not listed all)
We thanks the reviewer for all these improvements and changed the text accordingly
L. 21: "Stoichiometric changes"
L. 23-24: "as summer begins" --> "at the onset of summer"
L. 32: "to DIC"--> "with DIC"
L. 35: "a process named chemical carbonate compensation"
L. 50: "coastal waters"
L. 252: "in terms of"
Figure 4: panel (d) is not explained here.
L. 335: "significant"
L. 336: "no…were not performed" --> remove "not"